# Synergistic IL-6 and IL-8 paracrine signalling pathway infers a strategy to inhibit tumour cell migration

Hasini Jayatilaka[1], Pranay Tyle[1], Jonathan J. Chen[2], Minsuk Kwak[2], Julia Ju[1], Hyun Ji Kim[1], Jerry S.H. Lee[1,3], Pei-Hsun Wu[1,4], Daniele M. Gilkes[1,5], Rong Fan[2] & Denis Wirtz[1,4,5]

Following uncontrolled proliferation, a subset of primary tumour cells acquires additional traits/mutations to trigger phenotypic changes that enhance migration and are hypothesized to be the initiators of metastasis. This study reveals an adaptive mechanism that harnesses synergistic paracrine signalling via IL-6/8, which is amplified by cell proliferation and cell density, to directly promote cell migration. This effect occurs in metastatic human sarcoma and carcinoma cells– but not in normal or non-metastatic cancer cells-, and likely involves the downstream signalling of WASF3 and Arp2/3. The transcriptional phenotype of high-density cells that emerges due to proliferation resembles that of low-density cells treated with a combination of IL-6/8. Simultaneous inhibition of IL-6/8 receptors decreases the expression of WASF3 and Arp2/3 in a mouse xenograft model and reduces metastasis. This study reveals a potential mechanism that promotes tumour cell migration and infers a strategy to decrease metastatic capacity of tumour cells.

[1] Department of Chemical and Biomolecular Engineering, The Johns Hopkins University, Baltimore, Maryland 21218, USA. [2] Department of Biomedical Engineering, Yale University, New Haven, Connecticut 06520, USA. [3] Center for Strategic Scientific Initiatives, National Cancer Institute, Bethesda, Maryland 20850, USA. [4] Johns Hopkins Physical Sciences-Oncology Center, The Johns Hopkins University, Baltimore, Maryland 21218, USA. [5] Department of Oncology and Department of Pathology, Johns Hopkins University School of Medicine, Baltimore, Maryland 21231, USA. Correspondence and requests for materials should be addressed to D.M.G. (email: dgilkes1@jhu.edu) or to R.F. (email: rong.fan@yale.edu) or to D.W. (email: wirtz@jhu.edu).

Uncontrolled cell proliferation is a hallmark of cancer that leads to the development of primary tumours[1], which may be followed by further progression to metastasis, the spread of cancer cells from a primary organ to distal sites[2]. Cell proliferation and migration are two key drivers of metastasis that are regulated by complex interactions of multiple pathways that can either concurrently or divergently stimulate the two processes. Some studies have shown that the two processes occur simultaneously; proliferation and migration are both stimulated by secreted factors such as fibroblast growth factors[3,4]. Other studies suggest that the two processes are mutually exclusive; primary tumour cells proliferate uncontrollably with tight cell-cell junction and low mobility, while metastatic invasive tumour cells seem to delay proliferation during migration[5–9].

Cancer cells in the tumour microenvironment (TME) can secrete proteins, such as cytokines, that can interact with stromal and immune cells in a collagen-rich three-dimensional (3D) extracellular matrix[10,11], to mediate intercellular communications and collectively modulate pathophysiological processes, including cancer-induced angiogenesis and metastasis[12,13]. For instance, the highly invasive nature of human brain tumours, such as glioblastoma multiform, has been attributed to its unique secretomic profile[14]. However, secretomic profiles evolve as cancer cells proliferate and eventually progress to a higher grade (that is, as cells become more invasive)[15,16], suggesting a possible role for secreted paracrine factors to couple proliferation and migration. As the local concentration of secreted proteins increases with cell number, we speculate that the contribution of proliferation-induced local crowding, accompanied by increased local cell density in the collagen-rich 3D TME, may be a significant, yet largely unidentified factor that directly alters tumour cell migration[17].

In this study, we find that metastatic human sarcoma and carcinoma cells exhibit enhanced migration as a consequence of cell proliferation, which causes increased cell density in 3D collagen matrices. This increase in cell density causes significant enhancement in cell migration due to an increase in the secretion of specific soluble proteins. Using a high-throughput multiplexing cell secretomic profiling assay, we demonstrate that only interleukin 6 (IL-6) and Interleukin 8 (IL-8) are specifically increased with cell density. We also found that IL-6 and IL-8 are necessary and sufficient to increase tumour cell migration in a cell density dependent manner with negligible feedback on cell proliferation. This effect is specific to metastatic cancer cells; IL-6 and IL-8 have no effect on the migration of normal and non-metastatic cancer cells. Enhanced cell migration likely occurs through increased expression of Wiskott-Aldrich syndrome protein family member 3 (WASF3), which in turn regulates actin cytoskeleton dynamics through the recruitment of the Arp2/3 complex, increasing the formation of dendritic protrusions and thus driving cell migration[18]. Therapeutic targeting of the receptors of IL-6 and IL-8 using Tocilizumab and Reparixin significantly decreases the expression of the Arp2/3 complex in a mouse xenograft model and decreases metastasis of breast cancer cells to the lungs, liver, and lymph nodes. This study reveals a synergistic paracrine signalling pathway that when inhibited has the potential to decrease the metastatic capacity of cancer cells and thereby improve patient outcomes.

## Results

### Increase in tumour cell density enhances cell motility.
To assess the potential effect of cell proliferation–and associated increase in local cell density–on cancer cell migration *in vitro*, human fibrosarcoma HT1080 cells, a cell line commonly used in studies of cell migration[19–22], were embedded in 3D type I collagen matrices. Collagen I is not only the main extracellular matrix component of connective tissues, but is also enriched in the vicinity of carcinoma and sarcoma tumours[23]. Cell migratory patterns within the matrix were monitored for 16.5 h using live-cell phase-contrast microscopy at a rate of 30 frames/h every other day for 5 days. This analysis revealed that fibrosarcoma cells became progressively more motile as cells proliferated and increased local cell density (Fig. 1a–c). To investigate the role of increased cell density on cancer cell migration, we seeded increasing cell densities in 3D matrices and observed cell migration. The initial cell densities used in the experiments, ranging from 10 cells mm$^{-3}$ to 120 cells mm$^{-3}$, corresponded to average cell-to-cell distances from 470 to 130 µm in the 3D matrix, distances that were significantly larger than the average cell size (10–20 µm in diameter)[24]. This data revealed that cells became progressively more motile as cell density increased. This enhanced motility cannot be attributed to repulsive cell–cell interactions as the tracked cells did not come in contact with other cells. Cell speed eventually plateaued for cell densities higher than 100 cells mm$^{-3}$ (Fig. 1d,e). Similar trends were observed for migration parameters such persistence of migration and invasive distance (Fig. 1f and Supplementary Fig. 1A and B)[25].

We have previously shown that cell motility in 3D matrices is predicted by the ability of cells to form dendritic pseudopodial protrusions[19,26]. Consistent with these observations, we found that the total number of main and dendritic protrusions generated per unit time by tumour cells steadily increased and then plateaued with cell density. The cell-density-dependent number of protrusions generated by the cells is strongly correlated with the cell-density-dependent cell speed (Fig. 1g,h and Supplementary Fig. 1C).

This remarkable relationship between tumour cell density and cell migration was also found in human metastatic carcinoma cells (MDA-MB-231) and human metastatic glioblastoma cells (U-87). Similar to fibrosarcoma cells, the migration of these two tumorigenic, metastatic/invasive cell lines increased with cell density (Fig. 1i,j and Supplementary Fig. 1D). In contrast, cell-density-dependent migration was not observed in tumorigenic, non-metastatic carcinoma cells (MCF7) and non-tumorigenic cell lines WI-38 human lung fibroblasts and MCF10A human epithelial cells (Supplementary Fig. 1E–G).

Interestingly, cell-density-dependent migration was not observed when cells were placed on two-dimensional (2D) collagen-coated substrates (Supplementary Fig. 1H–J). Moreover, in contrast to 3D cell migration, the proliferation of cells in 3D matrices was unaffected by cell density (Fig. 1k and Supplementary Fig. 1K), that is, cells continued to proliferate at a constant rate regardless of cell density. Together, these results indicate that cancer cell density enhances cancer cell migration, but not proliferation, and that cell-density-dependent migration could be unique to tumorigenic, metastatic cells in 3D microenvironments.

### Cell migration enhancement is not caused by ECM remodelling.
Cell-density-dependent migration could be mediated by the collagen matrix and cell-induced matrix remodelling. We investigated if cell density modulated the microstructural properties of the matrix, such as inter-fibre spacing (effective pore size) and local fibre alignment[27]. Using quantitative reflection confocal microscopy, we determined that local fibre alignment showed poor correlation with cell density and speed. Average inter-fibre spacing showed a poor correlation with cell speed as well.

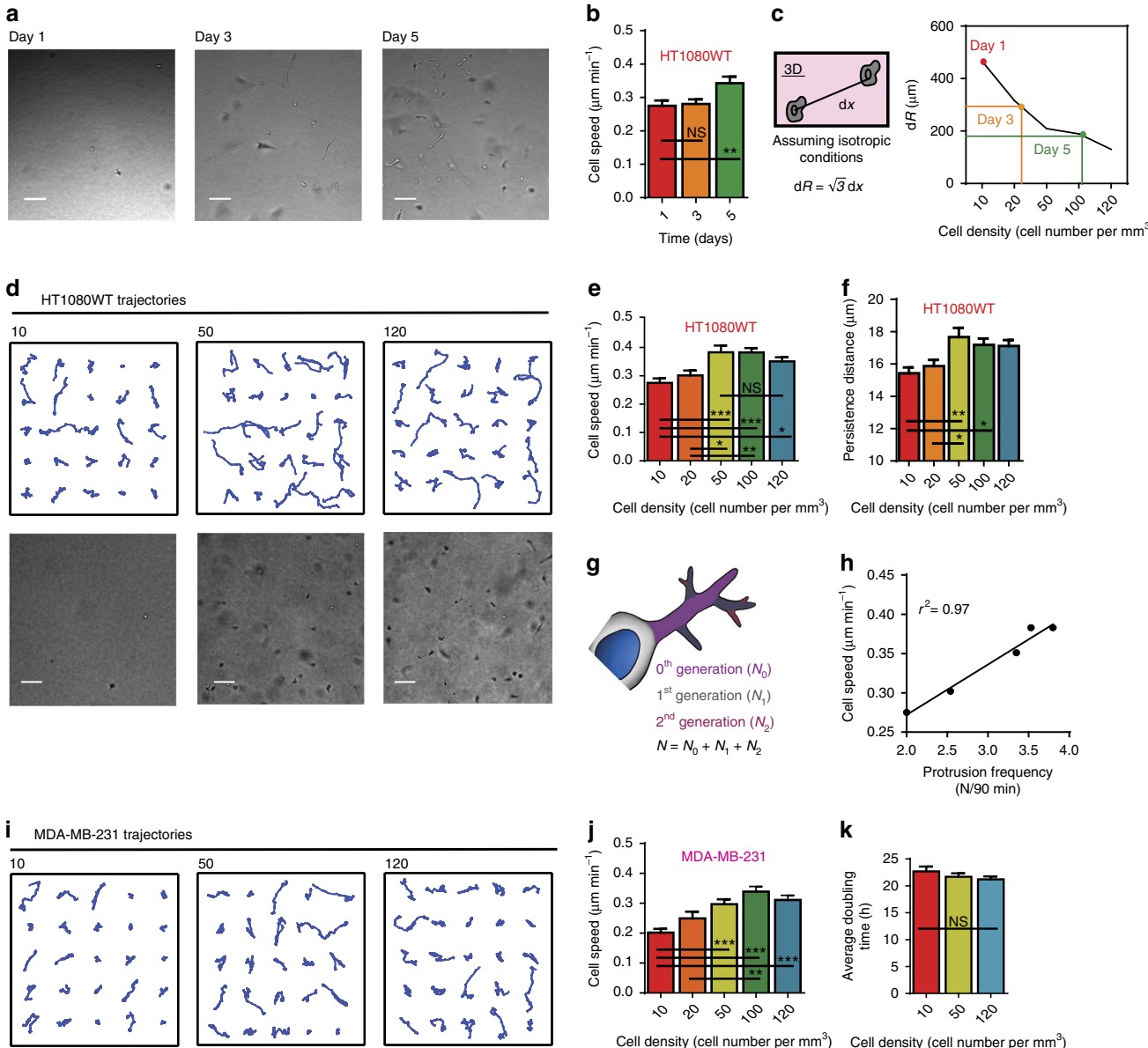

**Figure 1 | Effect of cell density on cancer cell motility.** (**a**) Phase contrast micrographs demonstrate confluence of human fibrosarcoma cells (HT1080WT) days after initial seeding. Scale bar, 100 μm. (**b**) Cell speed measured at a time lag of 2 min days after initial seeding. (**c**) Average distance to nearest cell (dR) relates density at different days to initial seeding density. (**d**) Randomly selected trajectories of human fibrosarcoma cells (HT1080WT) under different seeding densities of 10, 50, 120 cells mm$^{-3}$ embedded in a 3D collagen matrix. Phase contrast micrographs demonstrate the confluence at each density. Scale bar, 100 μm. (**e,f**) Cell speed and persistence distance measured at a time lag of 2 min at different seeding densities. (**g**) Topology of protrusions for cells embedded in 3D collagen matrices: 0th generation protrusions ($N_O$) originate from the cell body, 1st generation protrusions ($N_1$) stem from $N_0$ and 2nd generation protrusions ($N_2$) stem from $N_1$. (**h**) Cell speed and protrusion frequency are highly correlated. (**i**) Randomly selected trajectories of human carcinoma breast cancer cells (MDA-MB-231) under seeding densities of 10, 50, 120 cells mm$^{-3}$. (**j**) Cell speed evaluated at a time lag of 2 min, at five different seeding densities. Cells at high seeding densities ($\rho > 50$) show a significantly higher speed than cells seeded at low seeding density ($\rho = 10$). (**k**) Average doubling time at increasing cell density demonstrates that proliferation is independent of cell density. In all panels, data is represented as mean ± s.e.m. from three independent experiments. *$P < 0.05$; **$P < 0.01$; ***$P < 0.001$ (ANOVA) ($n = 3$).

As expected, a strong negative correlation was identified between average inter-fibre spacing and cell density. As the cell density increases, the forces exerted on the collagen fibrils by the cells increases, causing the space between the fibrils to decrease. (Fig. 2a–d and Supplementary Fig. 2A) Based on this result alone, we would have expected cell speed to decrease with increasing cell density but since the cells move faster as cell density increases this physical property cannot modulate cell-density-dependent migration. In sum, cell-density-dependent migration cannot be attributed to changes in the physical properties of the matrix.

**Secretomic profiles of matrix-embedded tumour cells.** Based on these results, we hypothesized that cell-density-dependent migration was regulated by soluble molecules secreted by the cells in a cell-density-dependent manner. To test this hypothesis, we introduced conditioned medium collected from a matrix containing a high density of HT1080 cells (50 cells mm$^{-3}$) into a matrix containing a low density of HT1080 cells (10 cells mm$^{-3}$). We found that the enhanced cell velocity observed at high cell density could be recapitulated at a low cell density by adding condition medium collected from high cell density matrices

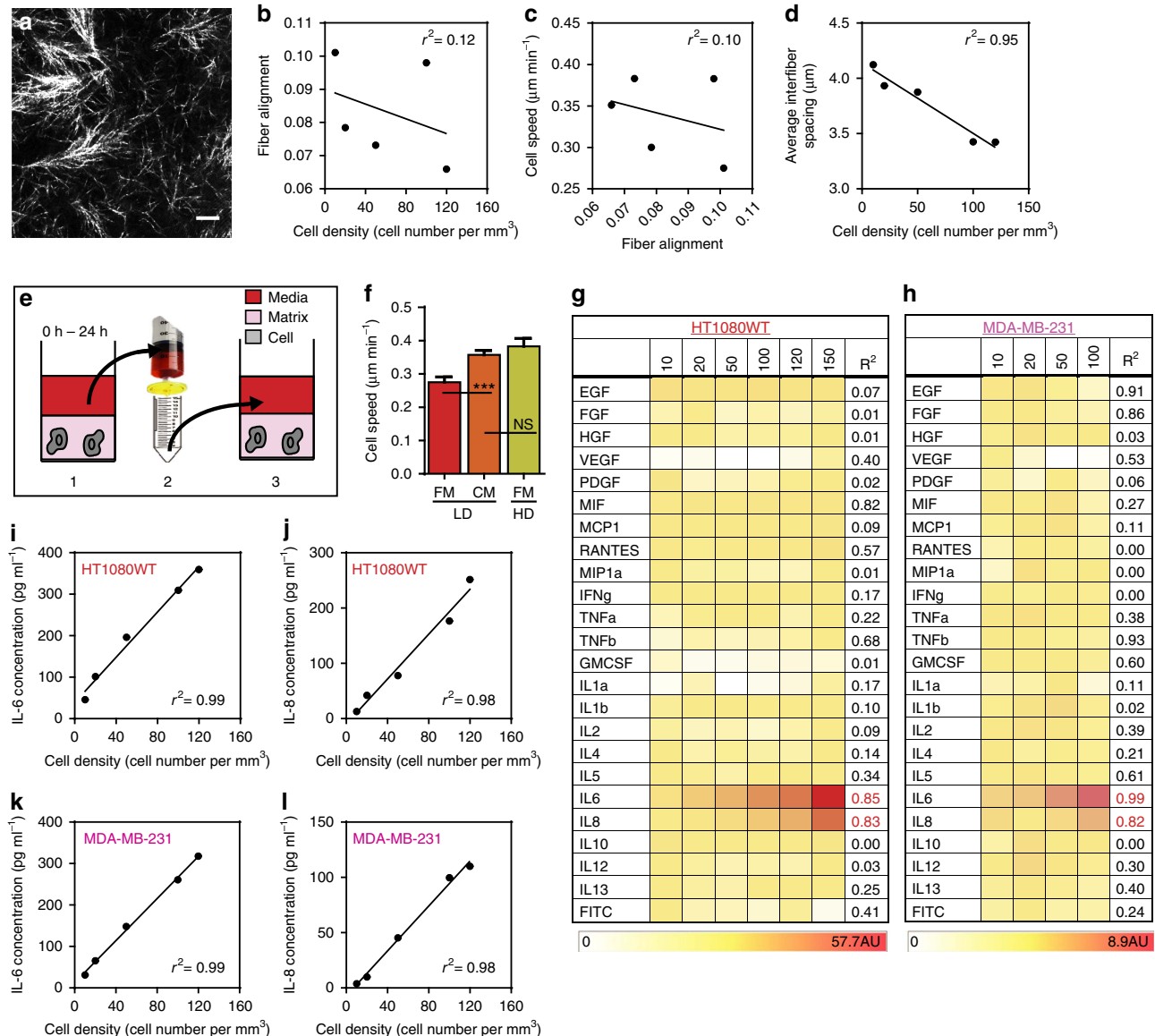

**Figure 2 | Biochemical cues.** (**a**) Reflection confocal micrograph. Singular of 3D collagen matrices. Scale bar, 10 μm. (**b**) Correlation plot of fibre alignment versus cell density. (**c**) Correlation plot of cell speed versus fibre alignment. (**d**) Correlation plot of average inter-fibre spacing versus cell density. (**e**) Method to prepare condition medium: medium is incubated for 24 h with a collagen matrix containing a high density of cells, 50 cells mm$^{-3}$ (HD), which is then filtered using a 0.45-μm filter, and added to a matrix containing a low density of cells, 10 cells mm$^{-3}$ (LD). (**f**) The addition of conditioned medium (CM) from a matrix containing a high cell density (HD) increases the speed of cells in a matrix containing a low cell density (LD). The HD cell speed in the presence of fresh medium (FM) is recapitulated in LD when using CM. (**g**) Secretomic analysis of CM harvested from human fibrosarcoma cells indicates that levels of interleukin 6 (IL-6) and interleukin 8 (IL-8) increase as a function of HT1080 cell density in the matrix, while levels of other major cytokines do not significantly change. (**h**) Secretomic analysis of conditioned medium from human breast carcinoma cells (MDA-MB-231) confirms our observations with HT1080 cells. (**i,j**) Increasing density of human fibrosarcoma cells in the matrix increases the concentrations of secreted IL-6 (A) and IL-8 (B), as analysed by ELISA. (**k,l**) Increasing cell density of human carcinoma cells in the matrix increases the concentrations of secreted IL-6 (A) and IL-8 (B), as analysed by ELISA. In all panels, data is represented as mean ± s.e.m. from three independent experiments. *$P < 0.05$; **$P < 0.01$; ***$P < 0.001$ (ANOVA).

(Fig. 2e,f). This result suggests that soluble molecules secreted by matrix embedded cancer cells are sufficient to promote enhanced cell migration.

To identify the soluble factor(s) driving enhanced motility, we measured and analysed the secretomic profiles of HT1080 and MDA-MB-231 cells embedded at low and high densities in 3D matrices, using a multiplex antibody microarray assay[28]. This assay simultaneously measured the concentration of 24 soluble molecules. We observed that the cytokines IL-6 and IL-8 were both secreted in relatively high concentrations and increased linearly with cell density for both cell lines. Remarkably, all other secreted proteins that were assayed including hepatocyte growth factor (HGF), which has been implicated in promoting tumour progression and tumour metastasis in several cancers[29], were not elevated at higher cell densities during our experimental time window (Fig. 2g,h). Using ELISA, we confirmed our results and determined the precise concentrations of IL-6 and IL-8 at specific cell densities of matrix-embedded HT1080 and MDA-MB-231 cells (Fig. 2i–l). Together, this result suggests that IL-6 and IL-8 drive density-dependent cell migration in 3D matrices.

**IL-6 and IL-8 together induce enhanced cell migration**. Next, we systematically assessed whether IL-6 and IL-8 were required to drive cell-density-enhanced migration by conducting gain-of-function and loss-of-function experiments. We exposed matrix embedded HT1080 cells seeded at a low density to controlled concentrations of human recombinant IL-6 and IL-8. We found that IL-6 or IL-8 alone had no effect on cell migration, even at high concentrations (Fig. 3a,b and Supplementary Fig. 3A and B). In contrast, IL-6 and IL-8, when combined at the prescribed concentrations found at the high density of 50 cells mm$^{-3}$ in the precise stoichiometric ratio of 5:2, induced cells at low density to move at the high velocity observed at high cell density and also detected for cells at low density exposed to conditioned medium (Fig. 3c). Strikingly, other stoichiometric ratios of IL-6 and IL-8 did not induce enhanced migration (Supplementary Fig. 3C). These results indicate that a mixture of IL-6 and IL-8 is sufficient to recapitulate the enhanced migration of cells embedded at high densities.

To verify that both cytokines were required for cell-density-dependent migration, we conducted experiments with conditioned medium from HT1080 cells depleted of IL-6 or IL-8 via shRNA interference. Depleting either IL-6 or IL-8 prevented the conditioned medium from high density matrices to enhance cell migration of low density matrices (Fig. 3d). Similar results were obtained when we utilized specific neutralizing antibodies to block secreted IL-6 and IL-8. The loss-of-function assays conducted with matrix-embedded cells at low and high cell densities exposed to specific neutralizing antibodies and with matrix embedded HT1080 cells depleted of IL-6 and IL-8 demonstrated that the cell-density-dependent migration patterns observed previously were no longer detected (Fig. 3e and Supplementary Fig. 3G–I). These results were confirmed with matrix embedded MDA-MB-231 cells (Fig. 3f). Interestingly, as with HT1080 cells, the enhanced migration of MDA-MB-231 cells was observed when IL-6 and IL-8 were present in the stoichiometric ratio of 5:2 (Supplementary Fig. 3D). In marked contrast, tumorigenic, non-metastatic cells, MCF7, and non-tumorigenic cells, MCF10A, exposed to both IL-6 and IL-8 did not exhibit enhanced migration (Supplementary Fig. 3E and F). These results suggest that IL-6 and IL-8 are each individually required, but only sufficient in combination to induce enhanced migration in tumorigenic, metastatic cells (Fig. 3g).

Further, we hypothesized that enhanced migration through the synergistic signalling of IL-6 and IL-8 is sensed by the cells via a paracrine pathway through the receptors of IL-6 (IL-6R) and IL-8 (IL8R1/CXCR1 or IL8R2/CXCR2). Matrix-embedded cells are exposed to a gradient of secreted proteins that can readily build up around a cell and consequently paracrine signalling can occur as the inter-cellular distance is decreased with an increase in cell density. As a result, the paracrine signalling could trigger a response in cellular behaviour (for example, enhanced migration). Our results demonstrated that the expression of IL-6R and CXCR2 indeed increased as cell density increased, indicating that signalling pathway is paracrine (Supplementary Fig. 3J and K).

We next explored possible therapeutic targets to decrease metastatic capacity by inhibiting the cell-density-dependent paracrine signalling pathway. The receptors on tumour cell membranes are also important drivers in signalling pathways and thus could be targeted to inhibit a particular phenotype. HT1080 and MDA-MB-231 cells depleted of IL-6R and CXCR2 (Referred to from this point on as IL-8R) via shRNA interference were embedded in 3D collagen matrices. We found that the depletion of IL-6R had no effect on cell migration at low cell density. In contrast, this molecular intervention suppressed cell velocity at elevated cell densities. Interestingly, cells depleted of IL-8R

displayed a reduction in cell velocity at both low and high cell densities (Fig. 3h and Supplementary Figs 3L and 6A–D).

To determine pharmacological agents for potential therapeutic interventions, inhibitors of IL-6R (Tocilizumab) and IL-8R (Reparixin) were added to matrix-embedded cells at low and high densities. Tocilizumab induced a small decrease in HT1080 cell velocity at low cell density, but induced a more visible decrease in cell velocity at high cell density. Reparixin decreased cell velocity at both low and high cell densities, with notable reduction in cell velocity at higher Reparixin concentrations (Fig. 3j). The combination of the two inhibitors showed a decrease in cell velocity at both low and high cell densities (Fig. 3k). We observed similar effects of the inhibitors on the velocity of MDA-MB-231 cells (Fig. 3l and Supplementary Fig. 3M and N).

**Cell density induces a distinct transcriptional phenotype**. To confirm the formation of a more invasive and migratory phenotype solely induced by cell proliferation and increase of cell density, we performed global transcriptional phenotype analysis by RNA sequencing (RNA-seq)[30]. The transcriptomes of HT1080 cells at low density (LD) and high density (HD) were sequenced and compared for differential gene expression. They were also compared with the transcriptomes of HT1080 cells at a low density exposed to recombinant IL-6 alone (IL-6), IL-8 alone (IL-8), and IL-6 and IL-8 found in the precise concentrations at the high density of 50 cells mm$^{-3}$ (RM). To identify the sources of transcriptional variations caused by different conditions, we performed an ANOVA-like test to detect the genes most variable among multiple groups. To study the relationship of global transcriptomes, principle component analysis (PCA) of the top 930 most significant genes was performed. PCA demonstrated that the transcriptomes of LD, IL-6 and IL-8 cluster in close proximity in the third quadrant while RM shows a phenotypical shift toward HD in the second quadrant. RM and HD residing in the same quadrant indicate the phenotypic similarity between cell-proliferation-induced migratory phenotype and the phenotype generated by RM (Fig. 4a). The shift in transcriptional phenotype induced solely by cell density increase was confirmed with differential gene expression analysis.

Next, we performed ingenuity pathway analysis (IPA) to investigate the biological mechanisms underlying transcriptional phenotypes by analysing the functional annotation of differential expression gene clusters and pathway enrichment. The most enriched gene ontology category of the genes highly expressed in RM was 'cell movement'. Another group of genes, which were highly expressed in IL-6, IL-8 and LD but downregulated in RM, was cellular metabolism and division-related pathways. This result suggests that RM induces a strong phenotype with enhanced cell movement. Other biological pathways contributing to the change of LD to HD cell phenotype include cell death and survival, cell metabolic activity, cell cycle and division. (Fig. 4b,c and Supplementary Fig. 4A and B).

**Mechanism of cell-density-dependent migration**. Signal transducer and activator of transcription 3, *STAT3*, is a transcription factor that is a common downstream effector in the individual pathways of IL-6 and IL-8 (refs 31,32). Therefore, we hypothesized that *STAT3* could regulate cell-density-dependent migration. We found that the activity of *STAT3* in matrix embedded HT1080 cells at a high density was two fold higher than that of cells at a low density (Fig. 4d).

Based on our previous work, we further speculated that the Arp2/3 complex nucleates F-actin assembly and mediates dendritic protrusions required for cell-density-dependent

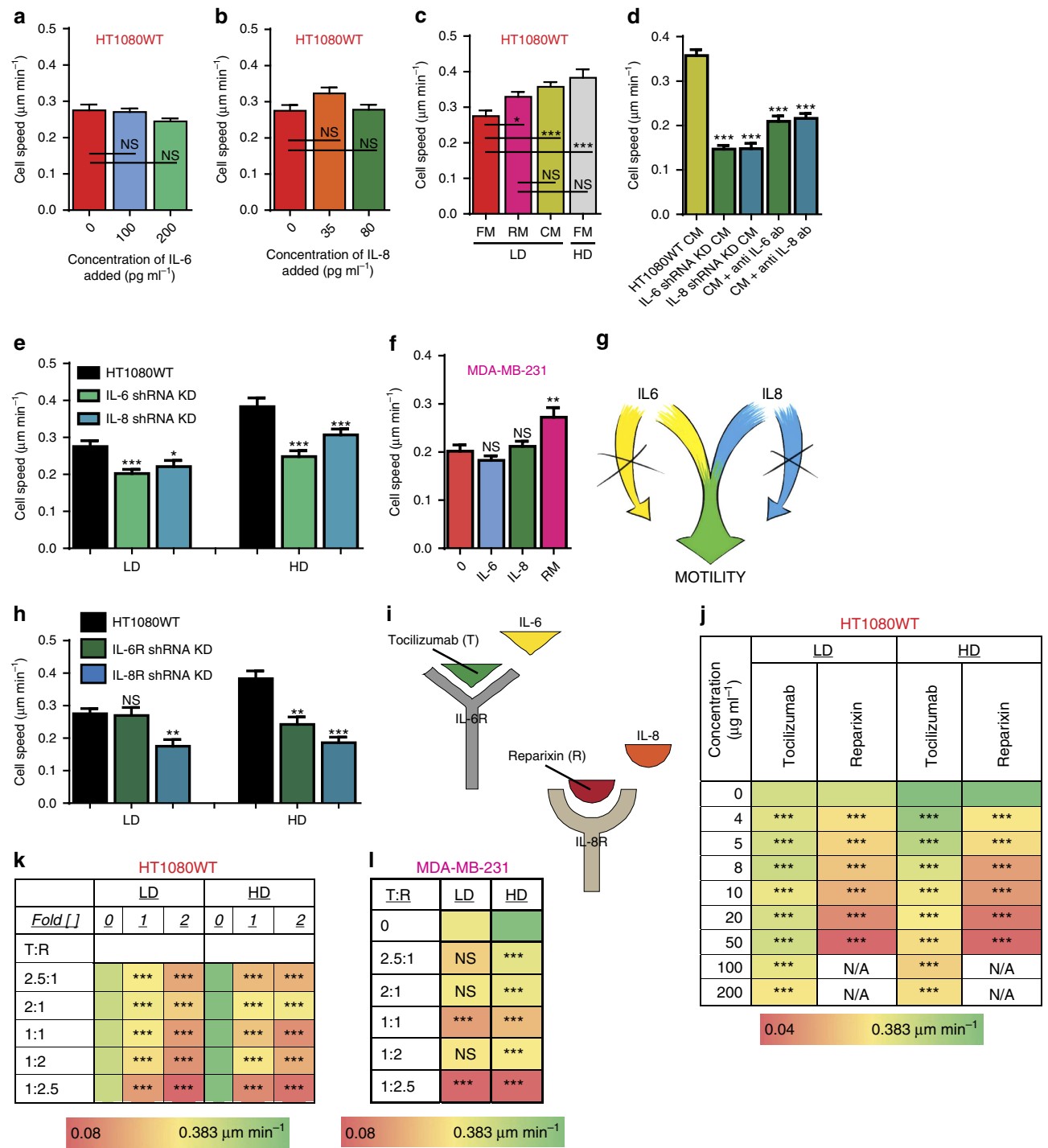

**Figure 3 | Functional influence of IL-6 and IL-8.** (**a**,**b**) The addition of recombinant IL-6 alone or recombinant IL-8 alone do not increase cell speed. (**c**) The addition of recombinant IL-6 and IL-8 in combination at the precise concentrations found in a matrix containing a high density of 50 cells mm$^{-3}$ (RM) recapitulates the high speed observed of human fibrosarcoma cells at high densities. (**d**) Decreased speed at LD ($\rho = 10$) where cells are exposed to conditioned medium produced by IL-6 and IL-8 knockdown cells and conditioned medium obtained from a matrix containing a high cell density (HD) following exposure to specific IL-6 and IL-8 functional antibodies compared with control cells exposed to conditioned medium from wild-type cells at HD ($\rho = 50$). (**e**) Decreased speed of the IL-6 and IL-8 knockdown cells at LD ($\rho = 10$) and HD ($\rho = 50$). (**f**) The addition of recombinant IL-6 and IL-8 in combination at the precise concentrations found in a matrix containing a high density of 100 cells mm$^{-3}$ recapitulates the high speed observed of human carcinoma cells at high densities. (**g**) Cartoon depicts the fact that IL-6 and IL-8 are both required to influence cancer cell motility. (**h**) Decreased speed of the IL-6R and IL-8R knockdown cells at LD ($\rho = 10$) and HD ($\rho = 50$). (**i**) Cartoon depicts that Tocilizumab and Reparixin can be used to block the cognate receptors of IL-6 and IL-8. (**j**) Individually, Tocilizumab and Reparixin decreased cell speed of human fibrosarcoma cells embedded in a 3D matrix at LD ($\rho = 10$) and HD ($\rho = 50$) compared with cells exposed to fresh medium (0). (**k**,**l**) Tocilizumab and Reparixin in combination greatly decrease cell speed of cells embedded in a 3D matrix at LD ($\rho = 10$) and HD ($\rho = 50$) compared with cells exposed to fresh medium (0). In all panels, data is represented as mean ± s.e.m. from three independent experiments. *$P < 0.05$; **$P < 0.01$; ***$P < 0.001$ (ANOVA).

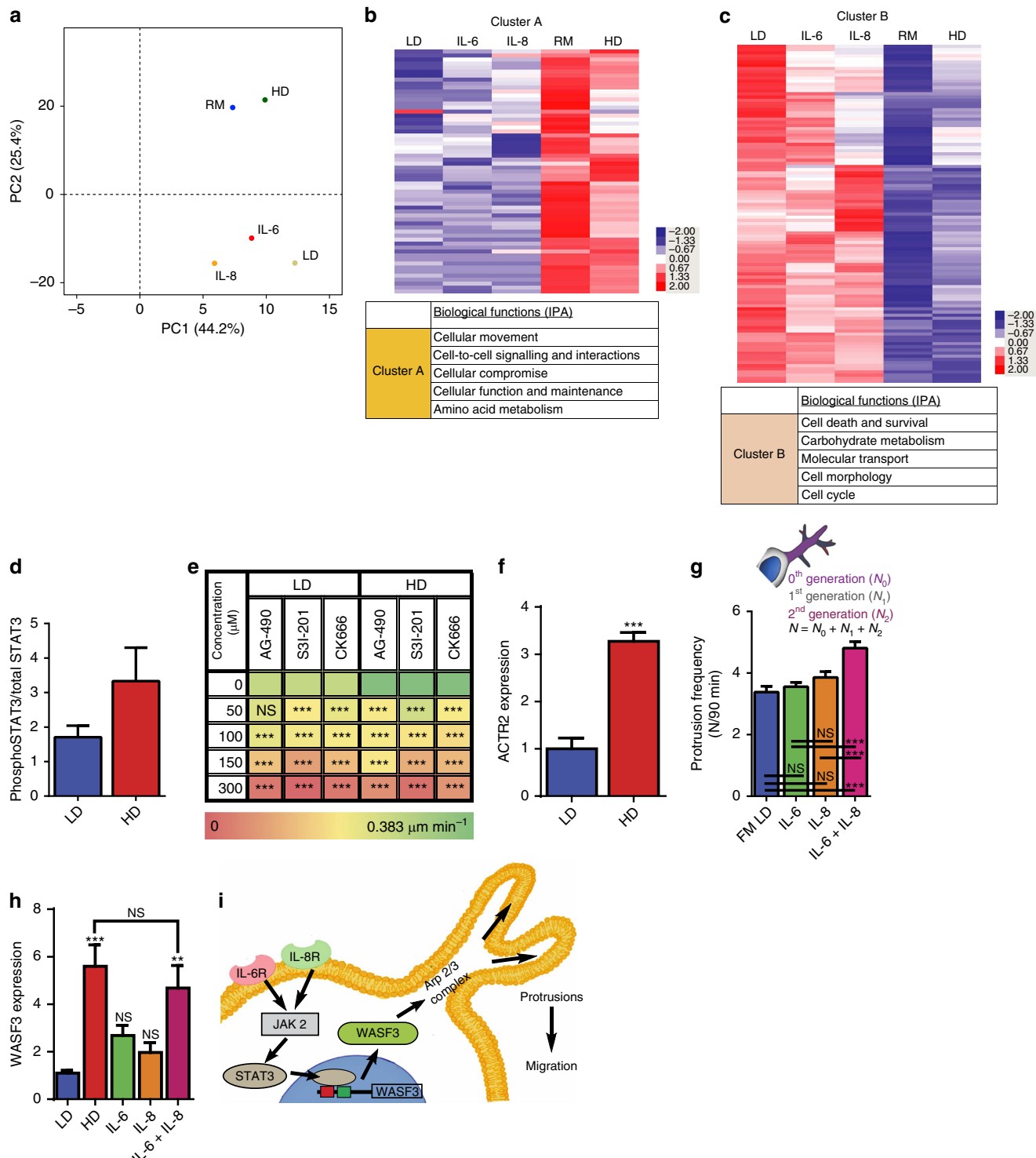

**Figure 4 | Proposed mechanism.** (**a**) Principle component analysis (PCA) of the top 930 most significant genes to determine the relationship of global transcriptomes. (**b**) Heat map demonstrating the difference between gene ontology categories. (**c**) Table demonstrating gene ontology categories. (**d**) Activity of STAT3 in 3D conditions at LD ($\rho = 10$) and HD ($\rho = 50$). (**e**) Decreased cell speed of human fibrosarcoma cells embedded in a 3D matrix exposed to JAK2 inhibitor, AG-490, STAT3 inhibitor, S3I-201, and Arp2/3 complex inhibitor, CK 666, at LD ($\rho = 10$) and HD ($\rho = 50$) compared with cells exposed to fresh medium (0). (**f**) Increased mRNA expression of ACTR2 at HD. (**g**) The addition of recombinant IL-6 and IL-8 alone does not increase protrusion frequency, however the addition of recombinant IL-6 and IL-8 in combination at the precise concentrations found in a matrix containing a high density of 50 cells mm$^{-3}$ (RM) significantly increases the frequency of cellular protrusions. (**h**) Increased mRNA expression of WASF3 at HD and cells exposed to recombinant IL-6 and IL-8 in combination at the precise concentrations found in a matrix containing a high density of 50 cells mm$^{-3}$. (**i**) Cartoon depiction of the IL-6 and the IL-8 signalling pathway leading to enhanced cell motility. In all panels, data is represented as mean ± s.e.m. from three independent experiments. *$P < 0.05$; **$P < 0.01$; ***$P < 0.001$ (ANOVA).

migration[19]. Thus, we reasoned that enhanced migration may be regulated by the Arp2/3 complex through the (Janus kinase) JAK/STAT3 pathway. Through examinations of the migration of HT1080 cells at low and high cell densities exposed to the specific JAK2 inhibitor AG-490 (ref. 33), STAT3 inhibitor S3I-201 (ref. 34), or Arp 2/3 complex inhibitor CK666 (ref. 35), we determined that JAK2, STAT3 and the Arp2/3 complex were indeed required for cell-density-dependent migration. Treatment with any of the three inhibitors prevented cell-density-dependent migration by repressing protrusion activity (Fig. 4e and Supplementary Fig. 4C).

To further determine the role of the Arp2/3 complex in cell-density-dependent migration, we measured the mRNA expression of the ACTR2 and protein expression of ARP2 and ARP3 and determined they wereslightly upregulated at HD (Fig. 4f and Supplementary Fig. 6E–H). We also measured protrusions and branching frequency for LD, HD, IL-6, IL-8 and RM conditions and demonstrated that IL-6 and IL-8 did not increase protrusion frequency or branching frequency but RM significantly did. (Fig. 4g and Supplementary Fig. 4D).

Because WASF3 is involved in the regulation of actin cytoskeleton dynamics through the recruitment of the Arp2/3 complex[36–38], we also hypothesized that WASF3 was an important intermediate between STAT3 and Arp2/3. Thus, we quantified the mRNA expression of WASF3 at LD, HD, IL-6, IL-8 and RM and found a relatively higher expression of WASF3 at HD and RM compared with LD, IL-6 and IL-8. Additionally, the protein level for WASF3 increased at HD (Supplementary Fig. 6I and J). RNAseq analysis also indicated that WASF3 was upregulated at RM but not when the interleukins were present individually (Fig. 4h and Supplementary Fig. 4E). Further we depleted WASF3 through shRNA interference and found that cell-density-dependent migration was not observed. (Supplementary Fig. 4F and G) These results suggest that WASF3 together with the Arp2/3 complex are important regulators in the pathway that controls cell-density-dependent migration (Fig. 4i).

Interestingly, when the expression of STAT3, WASF3 and ACTR2 were measured for differing stoichiometric ratios of IL-6 and IL-8, we observed that the expression of these three intermediates were maximally stimulated under the 5:2 conditions (Supplementary Fig. 4H–J).

**Mouse xenograft model to test therapeutic strategies**. Given that IL-6 and IL-8 cooperate to enhance migration in breast carcinoma cells, we sought to investigate their potential role in metastasis by inhibiting their cognate receptors using Tocilizumab and Reparixin. Tocilizumab is currently in clinical trials to study its efficacy against recurrent ovarian cancer[20], while Reparixin is being evaluated for safety, tolerability, pharmacokinetics, and to detect early signs of antitumour activity in breast cancer patients[22,23]. The effect of these drugs on metastasis was examined by generating an animal model through the introduction of MDA-MB-231 carcinoma breast cancer cells into the mammary fat pad of NSG (NOD SCID Gamma) mice and injecting four sets of mice with saline, Tocilizumab alone ($25\,mg\,kg^{-1}$), Reparixin alone ($30\,mg\,kg^{-1}$), and Tocilizumab and Reparixin in combination every three days for 6 weeks.

As predicted from our *in vitro* results (Fig. 1k), we observed that the treatment had no effect on the rate of tumour growth (Supplementary Fig. 5A and B). We also determined that metastasis to the lungs, liver, and lymph nodes were suppressed in the treated group. Specifically, the combination of the two drugs was the most effective in repressing metastatic burden on the liver and the lymph nodes (Fig. 5a–f). Moreover, the expression of the key intermediates in the synergistic pathway, STAT3, WASF3 and ACTR2, were significantly decreased in the treated group suggesting that the cell-density-dependent paracrine signalling pathway was responsible for the decreased metastases observed. (Fig. 5g–i) Immuno-histochemical staining for ACTR2 confirmed expression in control tumours and markedly decreased expression in the tumours from mice treated with the combination of drugs (Fig. 5j,k).

Our *in vitro* and *in vivo* findings describe a novel synergistic paracrine signalling pathway between IL-6 and IL-8 that plays a critical role in metastasis through the regulation of cell-density-dependent tumour cell migration. Thus, this study infers a new therapeutic target to decrease the metastatic capacity of tumour cells and improve patient outcomes.

**Discussion**

This study suggests that as cancer cells proliferate and local cell density increases accordingly, the secretion profile of cancer cells in the TME may be dynamically altered and this may play an important role in metastasis. Increased local cell density directly enhances cell migration in metastatic cells embedded in 3D matrices by increasing IL-6 and IL-8 levels. Interestingly, cell-density-dependent migration is unique to tumorigenic, metastatic cells exposed to 3D microenvironments, not 2D flat surfaces, which reconstitute features of tissues that enable *in vitro* recapitulation of *in vivo* function including spatiotemporal gradients of biochemical cues such as cytokines and growth factors[39].

Further, this study identified a novel synergistic mechanism between IL-6 and IL-8 required to promote cell-density-dependent migration. IL-6 is a pleiotropic cytokine that has been associated with tumour progression and metastasis in different types of cancers[40–42]. IL-8 has also been implicated in promoting angiogenesis and tumorigenicity, promoting a cancer stem cell phenotype, and enhancing metastasis in multiple cancer types[43,44]. Moreover, clinical data indicates that both IL-6 and IL-8 are found at high concentrations in the serum of patients with lung and liver metastases[45], which suggests that they may play a critical role in metastasis[46]. Previous clinical studies have also shown that the serum concentrations of these two cytokines strongly correlate with the stage of cancer[47]. The results of this study explain the underlying mechanisms that account for these observations in 3D collagen-rich TMEs where tumour cells rapidly proliferate and increase local cell density[16,48,49]. Strikingly, our data also suggests that this synergistic IL-6/8 paracrine pathway-mediated enhancement of cell migration is an adaptive process dictated by cell signalling and differs from the mechanism associated with cancer-stem-cell (CSC). The cells used in this investigation did not display CSC characteristics as revealed by bulk RNA-seq[50,51]

Our findings further emphasize the necessity of 3D cultures in pharmaceutical studies as monolayer cell culture methods remain the *de facto* prevalent testing platform[51]. Cells cultured on dishes adopt physiologically irrelevant morphology and signalling patterns[52]. For instance, the cell-density-dependent migration seen in 3D is not observed on 2D substrates. In addition, there are currently no commercial therapeutics that specifically target metastasis which is responsible for 90% of cancer related deaths[53]. This study suggests that by concurrently inhibiting the identified pathway with Tocilizumab (a humanized monoclonal antibody currently used to treat rheumatoid arthritis[54]) and Reparixin metastasis can be directly targeted and decreased. Although these agents are individually studied in clinical trials for their efficacy against different types of cancers[55–59], they have not been

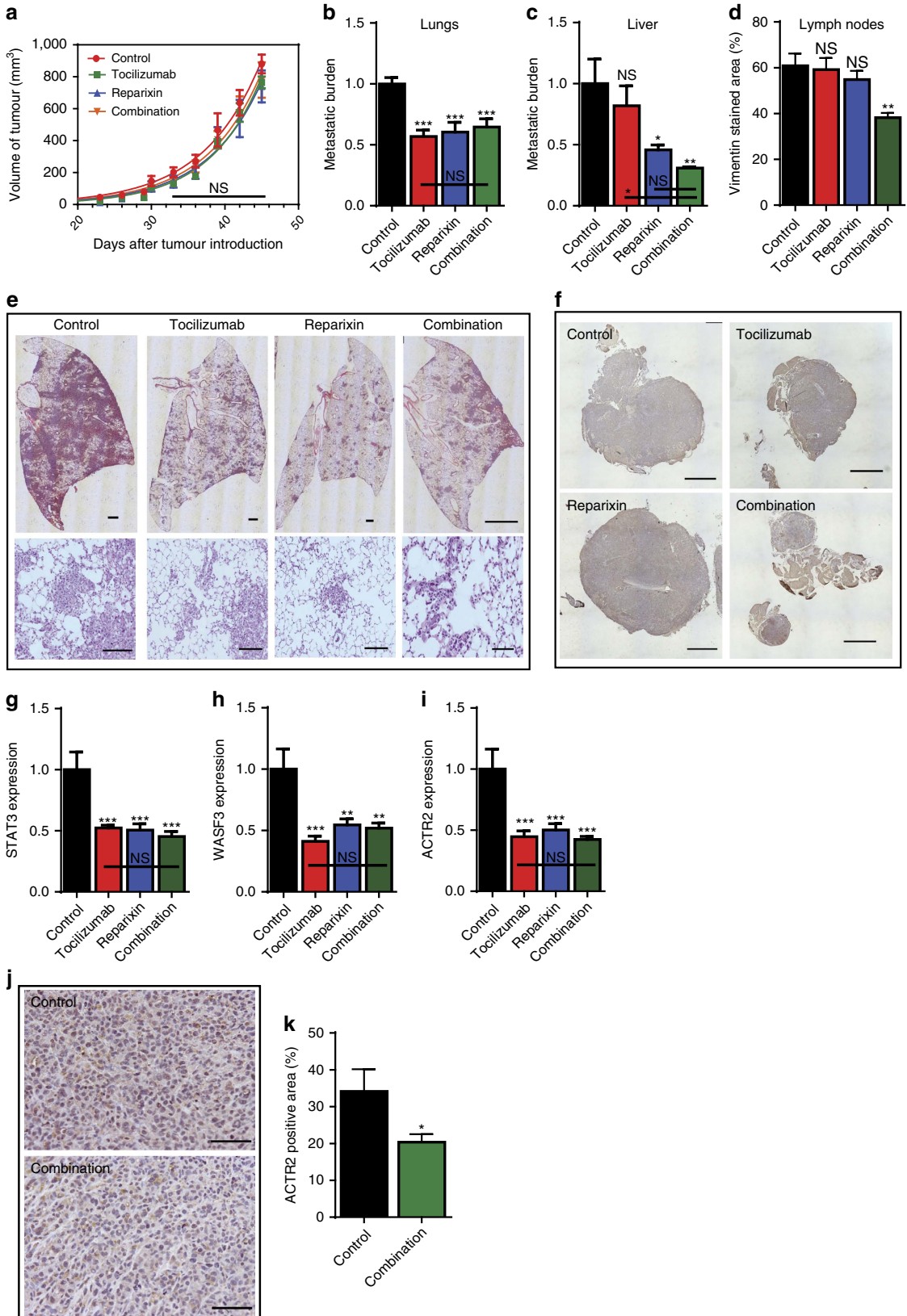

**Figure 5 | *In vivo* validation.** (**a**) Tumour volume measured over time. (**b,c**) Human genomic DNA content in mouse lungs and livers were quantified using qPCR to determine the metastatic burden. (**d**) Vimentin staining of lymph nodes quantified by image analysis. (**e**) Images of mice lungs that were stained with hematoxylin and eosin. Scale bar, 100 μm. (**f**) Images of lymph nodes that were stained with vimentin. Scale bar, 100 μm. (**g–i**) Decreased expression of STAT3, WASF3 and Arp2/3 in treated group compared with the control group. (**j**) Immunohistochemical staining of primary tumour sections for Arp2/3. Scale bar, 100 μm. (**k**) Arp 2/3 staining of primary tumour sections quantified by image analysis. In all panels, data is represented as mean ± s.e.m. of five mice. *P < 0.05; **P < 0.01; ***P < 0.001 (ANOVA).

evaluated as anti-metastasis therapeutics in highly invasive cancers. This study suggests that simultaneous use of Tocilizumab and Reparixin could greatly decrease the metastatic capacity of tumours, thereby potentially improve cancer patient outcomes similar to Denosumab which is a human monoclonal antibody that was initially developed to treat osteoporosis[60], but later found to be effective in the treatment of multiple myeloma and giant cell tumour of the bone[61].

Our observations also demonstrate that IL-6 and IL-8 work uniquely in a cell-autonomous manner through paracrine signalling amongst the same population of cancer cells. Previously, inflammatory signals released from infiltrating immune cells have been implicated in tumour progression, invasion, and metastasis. However, our work revealed a new mechanism for tumour-secreted IL-6 and IL-8, but not common inflammatory cytokines such as TNFα and IL-1β secreted by immune cells, to promote tumour cell migration and metastasis. Through this study, we have shown that the inhibition of the synergistic IL-6/8 signalling pathway reduces metastatic burden in mice, suggesting a potential strategy to prevent or treat cancer metastasis through the inhibition of tumour cell-density-dependent migration.

## Methods

**Cell culture.** Human fibrosarcoma HT1080 cells (ATCC) were cultured in Dulbecco's modified Eagle's medium (DMEM, Mediatech) supplemented with 10% (v/v) fetal bovine serum (FBS, Hyclone Laboratories), and 0.005% (w/v) gentamicin (Quality Biological). Human breast carcinoma MDA-MB-231 cells (ATCC) and MCF-7 cells (ATCC) were cultured in DMEM (Mediatech) supplemented with 10% FBS (Hyclone). Human glioblastoma U-87 MG cells (ATCC) were cultured in DMEM (Mediatech) supplemented with 10% FBS (Hyclone). Human diploid cell line, WI-38, (ATCC) were cultured in Eagle's minimal essential medium (EMEM, Mediatech) supplemented with 10% FBS (Hyclone). Human breast epithelial MCF10A cells (ATCC) and MCF12A cells (ATCC) were cultured in DMEM supplemented with 5% horse serum (Atlanta biologicals), 20 ng ml$^{-1}$ Human epidermal growth factor (Sigma-Aldrich), 100 ng ml$^{-1}$ cholera toxin, (Sigma-Aldrich) 0.01 mg ml$^{-1}$ bovine insulin (Life technologies), and 500 ng ml$^{-1}$ hydrocortisone (Sigma-Aldrich). HT1080 cells transfected with shRNAs (see below) were grown in medium containing 1 µg ml$^{-1}$ puromycin. The cells were maintained at 37 °C and 5% CO$_2$ in a humidified incubator during cell culture and during live-cell microscopy. All cell lines were tested for mycoplasma and deemed free of contamination.

**Depletion of proteins with shRNAs.** Target proteins were depleted by cotransfecting the shRNA construct (Supplementary Table 1) with two other packaging plasmids, pMD.G VSV-G and pCMVΔR8.91 (encoding Gag, Pol, Tat and Rev) using Lipofectamine 3000 (Invitrogen, Carlsbad, CA, USA). 293T cells at around 80% confluence were transfected with a mixture of 6 µg of lentiviral shRNA construct, 8 µg of pCMVΔR8.91, and 1 µg of pMD.G VSV-G. The conditioned medium containing the lentivirus was harvested 48 h after transfection and filtered through a 0.4-µm filter (Millipore, Billerica, MA, USA) to remove cell debris. For transduction, HT1080 cells were grown to 50–60% confluence in a 6-cm cell culture dish. Medium (2 ml) containing lentivirus was mixed with 1 ml of fresh medium containing protamine sulfate (final concentration 10 µg ml$^{-1}$) and added to HT1080 cells. After 8 h incubation, the medium containing the viruses was replaced with fresh medium containing 1 µg ml$^{-1}$ puromycin for selection. shRNA constructs targeting various genes were purchased from Sigma. (Taken from Giri et al.[19]).

After lentiviral-mediated transduction, Enzyme Linked Immunosorbent Assays (ELISA) (Supplementary Fig. 3H and I), PCR studies (Supplementary Fig. 4G), and western blots (Supplementary Fig. 6A–D) were performed and only shRNAs showing more than 85% knockdown were used for subsequent studies.

The transfected cells were embedded in type I collagen matrices and incubated overnight at 37 °C and 5% CO$_2$ in a humidified incubator. The conditioned media from the cells were collected and filtered through a 0.45-µm filter (Millipore) to remove cell debris. The total quantity of IL-6 and IL-8 produced by the cells were measured using Human quantikine ELISA kits (R&D systems). Data represented was obtained from three independent experiments.

**3D collagen I matrix.** HT1080 cells were embedded in 2 mg ml$^{-1}$ type I collagen gel as described previously by Fraley et al.[31] Briefly, cell suspensions containing 5,000 to 75,000 cells in 1:1 (v/v) ratio of cell culture media and reconstitution buffer were mixed with appropriate volume of soluble rat-tail collagen I (Corning Inc.) to obtain a final collagen I concentration of 2 mg ml$^{-1}$. A calculated amount of 1 M

NaOH was added quickly and the final solution was mixed well to bring the pH to ~7. The cell suspension was added to a 24-well coverslip-bottom cell-culture dish and immediately transferred to an incubator maintained at 37 °C to allow polymerization. Fresh medium was added 1 h before imaging. MDA-MB-231 and U-87 cells were embedded in 1 mg ml$^{-1}$ type I collagen matrix. The data represented for each cell line were obtained from three independent experiments.

**Speed and protrusion topology of matrix embedded cells.** Phase-contrast images of matrix-embedded cells were recorded 2 min apart for 16.5 h using a Cascade 1K CCD camera (Roper Scientific) mounted on a Nikon TE2000 microscope with a 10X objective lens. A minimum of 50 single cells were tracked using Metamorph imaging software. A custom MATLAB program calculated the velocity for each cell using the x- and y-coordinates obtained from tracking data using the following equation:

$$\text{Speed} = \frac{\sqrt{<[x(t+\Delta t) - x(t)]2 + [y(t+\Delta t) - y(t)]2>}}{t}.$$

For the characterization of protrusion topology, the movies were used to count the total number of mother protrusions, and the number of first-, second-, and third-generation protrusions generated by the cell (Fig. 1c). The protrusions emanating directly from the cell body, even when split, were termed mother protrusions; protrusions originating from the mother protrusions were termed first-generation, and so on. Mitotic cells were not included in the measurements. Persistence and invasive distance were obtained by the methods described by Wu et al.[30] Data represented was obtained from three independent experiments.

**Proliferation assays.** HT1080 cells were embedded in type I 3D collagen matrices in increasing cell numbers from 5,000 to 60,000 cells. Phase contrast images of the cells were recorded 8 min apart for 48 h. The average doubling time was obtained by measuring the time between the 1st and 2nd divisions. Cell viability assay using Prestoblue (Invitrogen) was also conducted on the matrix-embedded cells of increasing cell number. Fluorescence was measured every 6 h for 48 h. Data represented was obtained from three independent experiments.

**Collagen inter-fibre spacing and alignment.** Matrix embedded cells with cell densities ranging from 10 cells mm$^{-3}$ to 150 cells mm$^{-3}$ were imaged and analysed according to the methods highlighted in Fraley et al.[32] to determine the inter-fibre spacing and alignment. Data represented was obtained from three independent experiments.

**Condition medium and high throughput secretomic analysis.** Matrix-embedded cells with cell densities ranging from 10 cells mm$^{-3}$ to 150 cells mm$^{-3}$ were incubated for 24 h at 37 °C in a humidified incubator. The conditioned medium from the cells was then collected and filtered through a 0.45-µm filter (Millipore) to remove cell debris. High throughput secretomic analysis was conducted on the condition medium collected as described previously by Lu et al.[33] Data represented was obtained from three experimental replicates.

Conditioned medium from HT1080 cells embedded in matrices with a cell density of 50 cells mm$^{-3}$ was added to freshly made matrices with a cell density of 10 cells mm$^{-3}$. These conditions were replicated when extracting conditioned medium from the HT1080 transfected cells. Data represented was obtained from three independent experiments.

**Gain of function and antibody blocking assays.** Recombinant IL-6 and IL-8 (R&D systems) reconstituted in DPBS (Life technologies) were added to matrix embedded cells with a cell density of 10 cells mm$^{-3}$ and imaged as described above (See velocity and protrusion topology of matrix embedded cells). Matrix embedded cells with cell densities of 10 cells mm$^{-3}$ and 50 cells mm$^{-3}$ were exposed to specific IL-6 (Proteintech; catalog #21865-1-AP) and IL-8 (Proteintech; catalog #60141-2-Ig) functional antibodies, at a concentration of 0.5 µg ml$^{-1}$. Data represented was obtained from three independent experiments.

**STAT3 and WASF3 activity.** Matrix embedded cells with cell densities of 10 cells mm$^{-3}$ and 50 cells mm$^{-3}$ were incubated for 24 h at 37 °C in a humidified incubator. The matrices were exposed to cell lysis buffer and mechanically broken down using a syringe. The suspension was centrifuged and the supernatant was measured for STAT3 and PhosphoSTAT3 using an ELISA kit (Abcam). WASF3 expression was measured using qRT-PCR for the cell densities stated previously and matrix embedded cells with a cell density of 10 cells mm$^{-3}$ exposed to IL-6 and IL-8 alone and in combination at the precise concentrations found at a high cell density of 50 cells mm$^{-3}$. Total RNA isolation was performed with RNA MiniPrep kit (Zymo research). cDNA synthesis was carried out as previously described by Gilkes et al.[62] The sequence for the cDNA primers that were used during PCR are found in Supplementary Table 2. Data represented was obtained from three independent experiments.

**Transcriptome analysis by RNA sequencing and analysis of RNA-Seq data.** Total RNA isolation was performed as described in the previous section. cDNA synthesis, sequencing and analysis of RNA-Seq data were conducted as previously described by Guo et al.[63] Data represented was obtained from one experiment.

**Inhibitor assays.** Matrix embedded cells with low and high cell densities were exposed to IL-8R inhibitor, Reparixin (Cayman Chemical), IL-6R inhibitor, Tocilizumab (Genentech), JAK2 inhibitor, AG-490 (Santa Cruz Biotechnology), STAT3 inhibitor, S3I-201, (Santa Cruz Biotechnology) and Arp2/3 complex inhibitor, CK 666, (Sigma-Aldrich) for 1 h before cells were imaged as described above (see velocity and protrusion topology of matrix embedded cells). Cell viability assays using Prestoblue (Invitrogen) were also conducted on matrix embedded cells exposed to Reparixin and Tocilizumab. Data represented was obtained from three independent experiments.

**In vivo mouse work.** Studies using 5–7 week-old NSG (NOD SCID Gamma) mice were carried out according to protocols approved by the Johns Hopkins University Animal Care and Use Committee in accordance with the NIH Guide for the Care and Use of Laboratory Animals. All mice were housed at a temperature of 25 °C under a 12-hr dark/light cycle. NSG mice were obtained from Johns Hopkins Medical Institution. Tocilizumab and saline for injection were obtained from the research pharmacy of The Johns Hopkins Hospital. Reparixin was obtained from Cayman Chemical and Med Chem Express.

MDA-MB-231 cells were harvested by trypsinization, resuspended at $10^7$ cells ml$^{-1}$ in a 1:1 mix of PBS:Matrigel and $1 \times 10^6$ cells were injected into the mammary fat pad (MFP) of the mouse. After 10 days, mice received a subcutaneous injection of either 100 µl Tocilizumab alone (25 mg kg$^{-1}$), 300 µl Reparixin (30 mg kg$^{-1}$) alone or a combination of Tocilizumab and Reparixin. The control mice received 100 µl of a saline solution. Primary tumours were measured in two dimensions (a and b), and volume was calculated as $4/3\pi \, ((a \times b)/2)^3$. Mice were sacrificed after 6 weeks. Tumours were excised, weighed and processed for RNA isolation and tissue lysate preparation. Expression of STAT3, WASF3 and ACTR2 were measured using PCR methods.

Lungs were perfused with agarose. One lung was inflated for formalin fixation and paraffin embedding; the other lung was used to isolate genomic DNA for qPCR (Supplementary Table 2). Genomic DNA was isolated from the livers. The livers, tumours, and lymph nodes were sectioned and stained with Vimentin (Sigma; catalog #V2258) at a dilution of 1:200 and ACTR2 (Protein tech; catalog #10922-1-AP) at a dilution of 1:50. Images were acquired by Nikon Eclipse NI-U and analysed using ImageJ[64]. Data represented for each condition was obtained from five animals.

**Western blot.** Matrix embedded cells with cell densities of 10 cells mm$^{-3}$ and 50 cells mm$^{-3}$ were incubated for 24 h at 37 °C in a humidified incubator. Matrix embedded cells were lysed on ice in RIPA buffer, Protease inhibitor cocktail and Phosphatase-inhibitor (Roche Applied Science) and mechanically broken down using a syringe. The suspension was centrifuged and the supernatant was further centrifuged in Amicon Ultra Centrifugal Filters (Millipore) Protein concentration was measured by bradford assay (BioRad) and 50 µg of protein was loaded into each well. Proteins were separated on 10% SDS-polyacrylamide gels, and transferred onto polyvinylidene difluoride membranes (BioRad). These were blocked with 5% milk in TBS, then incubated with primary antibodies, CXCR2 (Novus Biologicals; catalog #NBP2-43810) (Dilution of 1:1,000), IL6R (Santa Cruz Biotechnology; catalog #sc-655) (Dilution of 1:1,000), and β actin (Cell Signaling Technology; #5125) (Dilution of 1:1,000). Membranes washed in 0.1% Tween-TBS and bound primary antibodies were detected with Peroxidase-conjugated goat anti-rabbit antibodies (Cell signaling technology) using the ECL detection system (BioRad) and a Chemdoc image analyzer (BioRad).

**Statistics.** The mean values ± s.e.m. were calculated and plotted using GraphPad Prism software (GraphPad Software). One-way ANOVA test was performed to determine statistical significance, which is indicated in the graphs using a Michelin grade scale ***$P < 0.001$, **$P < 0.01$ and *$P < 0.05$.

**Data availability.** The data that supports the findings of this study are available from the corresponding author on request. RNA-seq data can be found at NCBI GEO (GSE96817).

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

## Acknowledgements

This work was supported by National Cancer Institute grants (1U54CA210173-01, R01CA174388 and 5U54CA193461). We thank Max Matsuda Hirata, Fatima Umanzor, Adam Torres, Sierra Atwater, Aimee Dai, and Vishwesh Shah for their help with preparing samples and illustrations for the manuscript. We also thank Dr Kenneth J. Pienta, Dr Rong Li, Dr Konstantinos Konstantopoulos, and Dr Ie-Ming Shih for their constructive feedback.

## Author contributions

H.J., D.M.G., P.-H.W., R.F. and D.W. designed the experiments. H.J., J.J.C., M.K., H.J.K., P.T., J.J., and D.W. conducted the experiments and analysed the data. H.J. and D.W. wrote the paper. H.J., D.M.G., P.-H.W., J.S.H.L., M.K., D.W. and R.F. edited the paper.

## Additional information

**Competing interests:** The authors declare no competing financial interests.

