## [Peer Review File · Nature Communications]

Reviewers' comments:

Reviewer #1 (Remarks to the Author):

In this manuscript, the authors utilize a series of human sarcoma and breast carcinoma cell lines to elucidate a mechanism by which cellular density in three dimensions regulates cell mortality through pathways involving IL6, IL8, STAT3, WASF3 and APR2/3. In general, the experiments are well done and reported and reveal a novel mechanism linking cell density to cell mortality. However, there are a number of significant issues with the data as presented that need to be addressed.

1. In this manuscript, the authors assume that the cells within each of the cell lines are homogeneous with respect to IL6 and IL8 expression. In fact, other studies have demonstrated that this is not the case and that both the IL6 (Wang, "Stem Cells" 10: (2393), 2009) receptor and the IL8 receptor CXCR1 (Ginester, "Journal of Clinical Investigation" 120(2):485, 2010) are selectively expressed on stem like cells. Furthermore, there is considerable evidence that stem like cells which undergo an epithelial mesenchymal transition display increased mobility. The authors fail to account for these phenomenon by assuming that the effects of IL8 and IL6 are equivalent across the cell populations. To address this question, the authors would need to examine the effect of these cytokines on specific cell populations expressing IL6 receptor or the IL8 receptor CXCR1/2. In addition, the relationship between cellular invasion and EMT needs to be investigated and discussed.

2. Although the authors claim that their investigations elucidate relationships between cellular proliferation and mortality, in fact, all of the phenomena they study are directly related to cell density rather than cell proliferation. In fact, an important control would be to plate cells at high cell density in the absence of growth and determine if these cells still secrete IL6 and IL8. If that were found it would suggest that their phenomenon is completely density related and is not linked to proliferation.

3. The authors present interesting data demonstrating the importance of a precise stoichiometric ratio of IL6 to IL8 of 5 to 2. However, they provide no mechanistic insights as to why this ratio in particular increases cell mortality. It would be interesting to see if this ratio of cytokines maximally stimulates STAT3, WASF3 and APR2/3 signaling.

4. Figure 3H and 3J suggests that IL8 blockade effects cell migration at both low and high density.

Since IL8 is only secreted by cells grown at high density, how do the authors account for this effect?

5. Studies involving blockade of IL6 and IL8 signaling seem to demonstrate that simultaneous inhibition of both pathways maximally inhibits tumor metastasis to the liver and lymph nodes whereas each agents singly has equivalent effect to the combination in blocking lung metastases. How do the authors account for these discrepancies?

Reviewer #2 (Remarks to the Author):

In this manuscript by Jayatilaka et al., it is shown that sarcoma and carcinoma cells grown at high density in collagen gels display enhanced migration. This phenomenon requires IL6 and IL8 paracrine signalling. The importance of these findings is validated in xenograft assays, where inhibitors of IL6 and IL8 receptors decrease the metastatic burden.

The overall message is important as it points to a potential treatment of metastasis formation. The core of the demonstration centered on IL6 and IL8 is well demonstrated. There are already multiple evidence in the literature that IL6 and IL8 are involved in cancer progression and metastasis formation. But the way these factors were found in a logical scientific and unbiased line here makes this study interesting. However, some other parts, especially the one dealing with WASF3 and Arp2/3 are simply, of poor quality. I believe major revisions are required before this manuscript can be published in Nature communications.

1_I strongly disagree with the way the study is introduced in the abstract and introduction, by stating that proliferation and migration are mutually exclusive. The authors can give a couple of references in support of this claim. But just as many can be given in support of the opposite claim. The authors should think of growth factors that trigger motility AND proliferation. HGF/SF is probably the best example but this is actually a general scheme and it works the same with EGF or PDGF. Another example at the level of the Arp2/3 complex is the Arp2/3 inhibitory protein Arpin negatively regulate migration and proliferation in a coordinated manner. So the abstract statement that a direct coupling between migration and proliferation had never been observed is simply shocking. There is no need to build this artificial opposition between proliferation and migration to enter into this study.

2_I was expecting some discussion of how cell density is sensed in mammalian cells. It should be discussed and if there are some easy ways to show that the current observations are in a classical framework or on the contrary in an atypical one, they should also be performed. Paracrine signaling in response to cell density is the core of the paper.

3_Migration is not fully analyzed. Just speed is displayed. From the same trajectories the authors should extract the overall migration efficiency reflected in MSD and the directional persistence which together with speed determines the overall migration efficiency.

4_Experiments are displayed with error bars. We are never really explained what is plotted. Mean + or - sem ? How many technical replicates ? How many biological replicates ? Have these results been reproduced ever ? If yes, how many times ? All knock-downs are performed with a single shRNA sequence, which is not at all the standard. At least two independent sequences should give the same phenotype before a molecule can be demonstrated to have a role.

To anticipate an easy rebuttal : some results are antagonistic between shRNA and inhibitors with IL6 or IL8 also having a role at low cell density. So inhibitors cannot simply be used as independent confirmation of the shRNA effects. ShRNA results should be ascertained with two sequences. Sigma sells several sequences for each gene. This is straightforward to validate the presented results.

5_RNAseq : The authors claim that the main result of this analysis is that medium containing both IL6 and IL8 at the right stoichiometry recapitulates the high density medium. This is a very important point of the demonstration. But is it really the case ? The principal component analysis of results does not clearly indicate that HD and RM conditions are clustered together. HD is closer to the LD, IL6, IL8 conditions than to the RM with which the authors try to make it fit. There should be an objective score associated with this PCA. The visual impression on two clusters of selected genes does not suffice to demonstrate this point. Nowhere can I find details as to how these two clusters were defined. Nor what these genes having a biological function in 'cellular movements' are. Are WASF3 and Arp2/3 gene among them ? The RNAseq data should be deposited in an appropriate database (GEO) and the results concerning the clusters should be made available as supplementary data.

I guess WASF3 and Arp2/3 were simply selected as good candidates based on the extensive literature implicating them in migration. But the writing is ambiguous on this point. I also disagree with the statement that the RNAseq analysis suggests that RM induced a reduced cell metabolic activity, cell cycle and division. If this point is of importance for the manuscript, it should be directly demonstrated. Formulated as it is, it is just to mystify the reader to make him believe that migration should oppose proliferation (see first point).

6_Mechanism p10 : JAK2, STAT3, Arp2/3 inhibitors do not only affect density induced migration but also basal migration. This is absolutely expected for the Arp2/3 complex, which is well established to be required for protrusion based migration. Actually for me, the data with CK666 should be associated with Fig1F and G about the basic characterization of migration in this system. The fact that JAK2 and

STAT3 are required for basal migration is puzzling to me. How do the author interpret these results ? Do these results actually confirm the role of IL6 and IL8 in density induced migration ? I found them a bit problematic.

7_WASF3 is induced at the mRNA level. What about the protein ? WASF3 has been found to be critical for invasion by the groups of Sossey-Alaoui and Cowell and dispensable by Machesky's group, in the same cell line MDA-MB-231 used here. Since the authors went to suggest that WASF3 is part of the mechanism of migration and invasion, they should demonstrate that WASF3 is indeed required through shRNA mediated depletion.

8_Arp2/3. The Arp2/3 complex is a complex. Here one subunit only is measured at the mRNA level and conclusion is drawn for the whole complex. We don't even know which subunit is measured. On what criterium has this subunit been selected ? Is the level of the chosen subunit reflecting the level of others ? There are also paralogous subunits in Arp2/3 complexes, with different functions (recent work from Michael Way's group). Furthermore, one wonders why the chosen Arp2/3 subunit is not measured in Fig.4 like WASF3 for its induction by IL6 and IL8. The overall Arp2/3 complex should be measured at the protein level to conclude about any putative induction.

Arp2/3 staining in IHC is displayed in Fig5J. I cannot see well the staining on the figure... The positive area is nonetheless measured. Can the intensity be measured ? Is it varying ? What does the varying area means ? Intratumoral heterogeneity of samples ? If yes, what are the morphological structures that express the Arp2/3 ? The claimed data raise all these questions, but the basics are not ascertained. The exact reference of the Ab used is not given. Again, which subunit does it target ? Is it the same as the one which has been measured by qRT-PCR ? Has the antibody been validated for IHC ? How can we be sure that it binds to the expected target ?

Kaplan Meier of public Arp2/3 data. This is a strange way to finish the paper as the implication of the Arp2/3 as prognosis factor in many cancers has been published for a long time already. None of these references are cited, they should if the authors want to make this point that the Arp2/3 is a prognosis factor. They should not pretend to discover this point on public data: Otsubo et al., Mod Pathol 2004 ; Semba, Clin Can Res 2006, Iwaya et al. Mod Pathol 2007 ; Iwaya et al, Cancer Sci 2007 ; Zheng et al, Anticancer Res. 2008 ; Kashani-Sabet et al PNAS 2009 ; Liu et al, Oncol Report 2013.

9_ambiguities:

P5 l 107-108 : for 16.5 h or for 5 days ?

P8 l189 : shRNA induction ? I did not see that shRNA treatments were induced...

Response to reviewers' comments

Manuscript title: Novel synergistic IL6/8 paracrine signaling pathway infers new strategy to inhibit tumor cell migration

Authors: Hasini Jayatilaka, Pranay Tyle, Jonathan J. Chen, Minsuk Kwak, Julia A. Ju, Hyun Ji Kim, Jerry S.H. Lee, Pei-Hsun Wu, Daniele M. Gilkes, Rong Fan, and Denis Wirtz

We sincerely thank the reviewers for all their constructive criticism that has helped greatly improve the manuscript and the presentation of our work. The changes in the text have been highlight. Below is a summary of all the changes we made to the final manuscript.

1. Revised abstract and introduction to reflect reviewer 2's comments about proliferation and migration occurring both concurrently and divergently
2. Revised the results section to clarify the RNA seq data as per reviewer 2's comments.
3. Revised discussion section to address reviewer 1's comment that our observations may be associated with cancer stem like cells.
4. Included additional data demonstrating the maximal stimulation of important intermediates in the pathway with specific IL6:IL8 ratio of 5:2. Relevant discussion was also added to the main text.
5. Included additional data demonstrating that cell-density-dependent sensing occurs through the paracrine signaling via the expression of IL-6R and IL-8R. Relevant discussion was also added to the main text.
6. Included additional data demonstrating that migratory parameters such as persistence distance and invasive distance show similar trends as the cell speed used throughout the paper.
7. Included additional data that demonstrates the role of JAK/STAT in the regulation of cell-density dependent protrusion activity. Relevant discussion was also added to the main text.
8. Included additional data that indicates that WASF3 and ARPC2 are also upregulated at a protein level in high density conditions.
9. Validated IL-6 and IL-8 knockdowns using a second sequence for the genes and demonstrated that two separate sequences revealed the same phenotype.
10. Revised the methods section and the figure captions to describe data that is displayed and the specific methods used to generate the data.

Reviewer #1 (Remarks to the Author):

In this manuscript, the authors utilize a series of human sarcoma and breast carcinoma cell lines to elucidate a mechanism by which cellular density in three dimensions regulates cell mortality through pathways involving IL6, IL8, STAT3, WASF3 and APR2/3. In general, the experiments are well done and reported and reveal a novel mechanism linking cell density to cell mortality. However, there are a number of significant issues with the data as presented that need to be addressed.

1. In this manuscript, the authors assume that the cells within each of the cell lines are homogeneous with respect to IL6 and IL8 expression. In fact, other studies have demonstrated that this is not the case and that both the IL6 (Wang, "Stem Cells" 10: (2393), 2009) receptor and the IL8 receptor CXCR1 (Ginester, "Journal of Clinical Investigation" 120(2):485, 2010) are selectively expressed on stem like cells. Furthermore, there is considerable evidence that stem like cells which undergo an epithelial mesenchymal transition display increased mobility. The authors fail to account for these phenomenon by assuming that the effects of IL8 and IL6 are equivalent across the cell populations. To address this question, the authors would need to examine the effect of these cytokines on specific cell populations expressing IL6 receptor or the IL8 receptor CXCR1/2. In addition, the relationship between cellular invasion and EMT needs to be investigated and discussed.

Thank you for the insightful comments. Indeed, there are several reports that there may exist rare side populations in MDA-MB-231 that have cancer-stem-cell (CSC)-like characteristics (e.g., Kim et al., PLoS One. 2012;7(11):e50423). We believe our results are associated with the majority rather than rare CSC-like subsets as revealed by bulk RNA-seq that shows a population level transcriptional phenotype change. Our key finding that synergistic IL6/IL8 paracrine pathway-mediated enhancement of cell migration is an adaptive process dictated by cell signaling and differs from the mechanism associated with the CSC phenotype. In line with the results mentioned by the reviewer, the supplementary data in the Ginestier et al manuscript demonstrates that inhibition of IL-8 alone has no effect on the MDA cell line.

The reviewer's comment on EMT is highly relevant. In fact, the cell lines used in this study are isolated from metastatic sites and are classified as having a mesenchymal phenotype (loss of e-caherin, etc.). The key finding in this work is that even after EMT, the cells respond to cell-density dependent migration enhancement that probably occurs once the cells have detached from the primary tumor.

The following literature has been added to the revised manuscript and relevant discussion included in the main text. **(New Citation #64 and #65)**

1. Ginestier, C., Liu, S., Diebel, M. E., Korkaya, H., Luo, M., Brown, M., ... & Guan, J. L. (2010). CXCR1 blockade selectively targets human breast cancer stem cells in vitro and in xenografts. *The Journal of clinical investigation*, 120(2), 485-497.

2. Kim, S. Y., Rhee, J. G., Song, X., Prochownik, E. V., Spitz, D. R., & Lee, Y. J. (2012). Breast cancer stem cell-like cells are more sensitive to ionizing radiation than non-stem cells: role of ATM. *PLoS One*, 7(11), e50423.

2. Although the authors claim that their investigations elucidate relationships between cellular proliferation and mortality, in fact, all of the phenomenons they study are directly related to cell density rather than cell proliferation. In fact, an important control would be to plate cells at high cell density in the absence of growth and determine if these cells still secrete IL6 and IL8. If that were found it would suggest that their phenomenon is completely density related and is not linked to proliferation.

We apologize for the lack of clarity when explaining the coupling of proliferation and migration. We recognize that the coupling of proliferation and migration is a consequence of cell density and therefore have made changes to the manuscript that reflects this. We hope to convey to the reader that as cells proliferate and local cell density increases, cancer cell migration is enhanced through a synergistic paracrine signaling pathway. Importantly, our experimental condition relies on cells being plated at sub confluence and thus the results differ from the effect of confluency on cell migration: (1) cells are still many cell lengths away from each other when this cell-density effect on migration is triggered and (2) we don't observe this effect on conventional flat dishes, only in 3D tissue-like environments (collagen matrices). Although recently studies have shown contact inhibition could unexpectedly promote the spreading of loose cells in tumor (Lin et al., *Nature Communications*, 2015 Apr 8;6:6619), single-cell time lapse tracking microscopy showed that our system does not involve significant cell-cell contact even at high cell densities tested in our study and provides a new mechanism never reported before.

Additionally, the expression of mKi67, a marker for cell proliferation, showed negligible decrease from low density to high density conditions (see below), further elucidating that enhanced migration is a consequence of paracrine signaling and cell density, but not related to proliferation.

Again, we appreciate the reviewer's comments and have revised the manuscript to clarify the relationship between cell density and proliferation. This phenomenon is mechanistically associated with intercellular signaling and cell density dependent, but the relevance in tumorigenesis *in vivo* turns out to be a consequence of cell proliferation.

1. Lin, B., Yin, T., Wu, Y. I., Inoue, T., & Levchenko, A. (2015). Interplay between chemotaxis and contact inhibition of locomotion determines exploratory cell migration. *Nature communications*, 6.

3. The authors present interesting data demonstrating the importance of a precise stoichiometric ratio of IL6 to IL8 of 5 to 2. However, they provide no mechanistic insights as to why this ratio in particular increases cell mortality. It would be interesting to see if this ratio of cytokines maximally stimulates STAT3, WASF3 and APR2/3 signaling.

Following the reviewer’s recommendation, we conducted experiments to determine the expression of STAT3, WASF3, and Arpc2 (ARP2/3) with the stoichiometric ratios of IL-6 and IL-8 of 5.8:2, 5:2, and 3.4:2. PCR data shown below demonstrates that STAT3, WASF3, and ARPC2 are maximally stimulated at the ratio of 5:2 of IL-6: IL-8. The following figures have been added to the revised manuscript and relevant discussion included in the main text. (Supplemental Figure 4I-K)

4. Figure 3H and 3J suggests that IL8 blockade effects cell migration at both low and high density. Since IL8 is only secreted by cells grown at high density, how do the authors account for this effect?

We apologize for any confusion in the representation of the data. IL-8 is in fact secreted by cells at a low density. Cells at a low density of 10 cells/mm³ secrete an average concentration of IL-8 of 12.8pg/mL as measured by ELISA with three independent experiments and three replicates each. Decrease in cell migration at low and high density during the blocking of IL-8 suggests that IL-8 plays an important role in cell-density dependent migration in both low and high densities conditions. But the enhanced migration requires a synergistic effect of IL6/IL8.

5. Studies involving blockade of IL6 and IL8 signaling seem to demonstrate that simultaneous inhibition of both pathways maximally inhibits tumor metastasis to the liver and lymph nodes whereas each agents singly has equivalent effect to the combination in blocking lung metastases. How do the authors account for these discrepancies?

Studies have shown that the first site of metastasis in both mice and humans is usually in the axillary lymph nodes. We find the observation that the metastatic burden in the lung does not change between the treated conditions very interesting. The discrepancies might be attributed to the difference between invasion (cell migration out of the primary tumor site) and metastasis (cells land to a distant site to establish a colony).

Metastasis involves a seed-and-soil hypothesis where the heterogeneous cell populations in the lung that consist of fibroblasts, monocytes, plasma cells, macrophages, polymorphonuclear leukocytes, pericytes, and alveolar macrophages can create a microenvironment that is favorable ('soil') for the circulating tumor cells to 'seed'.

If we were to think about the favorable conditions in terms expression of IL-6 and IL-8, we know that these cells, specifically fibroblasts, macrophages, monocytes, and leukocytes, produce these cytokines that can promote the establishment of secondary tumors. For instance, studies have shown that expression of IL-6 in specific organs such as the lungs, liver, or brain will attract circulating tumor cells to these organs and promote their establishment. IL-8 secreted from heterogeneous cell population is also shown to activate many signaling pathways that can promote an angiogenic response, inducing proliferation and survival. Additionally, studies have shown that both IL-6 and IL-8 play a role in seeding of circulating tumor cells.

Nevertheless, we show in this manuscript that some tumor cells can increase their migration autonomously without signals from the "soil", representing a new mechanism contributing to tumor invasion. We believe the relation between tumor cells and stromal environments requires further study that exceeds the scope of this paper.

1. Weigelt, B., Peterse, J. L., & Van't Veer, L. J. (2005). Breast cancer metastasis: markers and models. *Nature reviews cancer*, 5(8), 591-602.
2. De Larco, J. E., Wuertz, B. R., Rosner, K. A., Erickson, S. A., Gamache, D. E., Manivel, J. C., & Furcht, L. T. (2001). A potential role for interleukin-8 in the metastatic phenotype of breast carcinoma cells. *The American journal of pathology*, 158(2), 639-646.
3. Crapo, J. D., Barry, B. E., Gehr, P., Bachofen, M., & Weibel, E. R. (1982). Cell Number and Cell Characteristics of the Normal Human Lung 1–3. *American Review of Respiratory Disease*, 126(2), 332-337.
4. Waugh, D. J., & Wilson, C. (2008). The interleukin-8 pathway in cancer. *Clinical cancer research*, 14(21), 6735-6741.
5. Ara, T., & DeClerck, Y. A. (2010). Interleukin-6 in bone metastasis and cancer progression. *European journal of cancer*, 46(7), 1223-1231.
6. Psaila, Bethan, and David Lyden. "The metastatic niche: adapting the foreign soil." *Nature Reviews Cancer* 9.4 (2009): 285-293.
7. Kim, M. Y., Oskarsson, T., Acharyya, S., Nguyen, D. X., Zhang, X. H. F., Norton, L., & Massagué, J. (2009). Tumor self-seeding by circulating cancer cells. *Cell*, 139(7), 1315-1326.

Reviewer #2 (Remarks to the Author):

In this manuscript by Jayatilaka et al., it is shown that sarcoma and carcinoma cells grown at high density in collagen gels display enhanced migration. This phenomenon requires IL6 and IL8 paracrine signaling. The importance of these findings is validated in xenograft assays, where inhibitors of IL6 and IL8 receptors decrease the metastatic burden.

The overall message is important as it points to a potential treatment of metastasis formation. The core of the demonstration centered on IL6 and IL8 is well demonstrated. There are already multiple evidence in the literature that IL6 and IL8 are involved in cancer progression and metastasis formation. But the way these factors were found in a logical scientific and unbiased line here makes this study interesting. However, some other parts, especially the one dealing with WASF3 and Arp2/3 are simply, of poor quality. I believe major revisions are required before this manuscript can be published in *Nature communications*.

I strongly disagree with the way the study is introduced in the abstract and introduction, by stating that proliferation and migration are mutually exclusive. The authors can give a couple of references in support of this claim. But just as many can be given in support of the opposite claim. The authors should think of growth factors that trigger motility AND proliferation. HGF/SF is probably the best example but this is actually a general scheme and it works the same with EGF or PDGF. Another example at the level of the Arp2/3 complex is the Arp2/3 inhibitory protein Arpin negatively regulate migration and proliferation in a coordinated manner. So the abstract statement that a direct coupling between migration and proliferation had never been

observed is simply shocking. There is no need to build this artificial opposition between proliferation and migration to enter into this study.

We apologize for the lack of clarity when describing proliferation and migration as mutually exclusive. The concept of historic “go-or-grow” hypothesis is based on the observation of cell behavior at a given time point (see references 1-6 below). In a prolonged timeframe, a cell may take a sequence of events to realize proliferation and migration/spreading. In addition, as the reviewer point out, it is often required physiologically since an external cue such as growth factor can trigger both motility AND proliferation although the “go-or-grow” concept is still valid; when the cell divides it does not migrate and when it migrates it is not dividing. The key finding of this work is that as cells proliferate and local cell density increases, the effect of intercellular paracrine signaling is enhanced, leading to secretion profile change and influencing an enhancement of cell migration, which is implicated in tumor invasion and metastasis. We believe the way of coupling of cancer cell proliferation and migration through cell density discovered in this work is novel. We have made changes to the manuscript to convey this message more clearly. We thank the reviewer for the valuable feedback. (New citations #4, #5, and #10)

1. De Donatis, A., Ranaldi, F., & Cirri, P. (2010). Reciprocal control of cell proliferation and migration. *Cell Communication and Signaling*, 8(1), 20.
2. Evdokimova, V., Tognon, C., Ng, T., & Sorensen, P. H. Reduced proliferation and enhanced migration: two sides of the same coin? Molecular mechanisms of metastatic progression by YB-1. *Cell cycle*, 8(18), 2901-2906, (2009)
3. Palmqvist, R., Rutegård, J. N., Bozoky, B., Landberg, G., & Stenling, R., Human colorectal cancers with an intact p16/cyclin D1/pRb pathway have up-regulated p16 expression and decreased proliferation in small invasive tumor clusters. *The American journal of pathology*, 157(6), 1947-1953, (2000)
4. Svensson, S., Nilsson, K., Ringberg, A., & Landberg, G., Invade or proliferate? Two contrasting events in malignant behavior governed by p16INK4a and an intact Rb pathway illustrated by a model system of basal cell carcinoma. *Cancer research*, 63(8), 1737-1742. (2003).
5. Hoek, K. S., Eichhoff, O. M., Schlegel, N. C., Döbbling, U., Kobert, N., Schaerer, L., ... & Dummer, R., In vivo switching of human melanoma cells between proliferative and invasive states. *Cancer research*, 68(3), 650-656, (2008).
6. Zheng, P. P., Severijnen, L. A., van der Weiden, M., Willemsen, R., & Kros, J. M. (2009). Cell proliferation and migration are mutually exclusive cellular phenomena in vivo: implications for cancer therapeutic strategies. *Cell Cycle*, 8(6), 950-951.

2_I was expecting some discussion of how cell density is sensed in mammalian cells. It should be discussed and if there are some easy ways to show that the current observations are in a classical framework or on the contrary in an atypical one, they should also be performed. Paracrine signaling in response to cell density is the core of the paper.

Thank you for the comments. The cell density sensing is presumably via paracrine signaling through IL-6 receptor (IL6R) and IL-8 receptor (IL8R1/CXCR1 or IL8R2/CXCR2) in a 3D matrix, in which a gradient of secretion proteins can readily build up around a cell and the paracrine signal occurs when the inter-cellular distance is decreased with the increase of cell density. As a result, the paracrine signaling could trigger a response of cellular behavior (e.g., enhanced migration) or the production of a second wave of cytokines to modulate cell density. This is the mechanism we believe how cell density is sensed in our study (mammalian cells).

This work focuses on the former effect – paracrine-induced cell behavior change (migration) and a systematic study was performed to measure the consequence of paracrine-induced intracellular signaling cascades linked to cytoskeleton, cell adhesion, and migration. In this work, we have measured both IL-6/IL-8 secretion and the expression of IL6R and IL8R (CXCR2) (See figure below), together with the downstream intracellular pathways leading to cell migration enhancement, which is a relatively complete study in the former aspect.

The latter is a complex systems biology and cell signaling research project similar as one of our recent publications (Xue et al., *Science Signaling*, 8(381):ra59, 2015) and is indeed one of our future directions. We have included the data below to the revised manuscript and added more discussion on these aspects. (Supplementary figure 3L,M)

1. Xue, Q., Lu, Y., Eisele, M.R., Sulistijo, E.S., Khan, N., Fan, R., Miller-Jensen, K., (2015). Analysis of single-cell cytokine secretion reveals a role for paracrine signaling in coordinating macrophage responses to TLR4 stimulation. *Science Signalling*, 8(381):ra59
2. Friedl, P., & Gilmour, D. (2009). Collective cell migration in morphogenesis, regeneration and cancer. *Nature reviews Molecular cell biology*, 10(7), 445-457.

3. Mayor, R., & Etienne-Manneville, S. (2016). The front and rear of collective cell migration. *Nature Reviews Molecular Cell Biology*.
4. Waters, C. M., & Bassler, B. L. (2005). Quorum sensing: cell-to-cell communication in bacteria. *Annu. Rev. Cell Dev. Biol.*, 21, 319-346.
5. Daniels, R., Vanderleyden, J., & Michiels, J. (2004). Quorum sensing and swarming migration in bacteria. *FEMS microbiology reviews*, 28(3), 261-289.
6. Redzic, J. S., Balaj, L., van der Vos, K. E., & Breakefield, X. O. (2014, October). Extracellular RNA mediates and marks cancer progression. *In Seminars in cancer biology* (Vol. 28, pp. 14-23). Academic Press.

3_Migration is not fully analyzed. Just speed is displayed. From the same trajectories the authors should extract the overall migration efficiency reflected in MSD and the directional persistence which together with speed determines the overall migration efficiency.

The cell speeds shown in the paper were obtained calculating the MSDs and then extracting the speed from it. Therefore cell speeds directly correlate to the MSDs. (Refer figure below) Following the reviewer's recommendation, we looked at persistence distance and invasive distance (as highlighted by Wu et al) of the matrix embedded cells with increasing cell density and found that the cells are more persistent at a high density than a low density. (Refer figure below) Following the reviewer's recommendation, this data was included in the revised manuscript. (Figure 1F and Supplementary Figure 1A and B) **(New citations #30).**

1. Wu, P. H., Giri, A., Sun, S. X., & Wirtz, D. (2014). Three-dimensional cell migration does not follow a random walk. *Proceedings of the National Academy of Sciences*, 111(11), 3949-3954.
2. Wu, Pei-Hsun, Anjil Giri, and Denis Wirtz. "Statistical analysis of cell migration in 3D using the anisotropic persistent random walk model." *Nature protocols* 10.3 (2015): 517-527.

4_Experiments are displayed with error bars. We are never really explained what is plotted. Mean + or – sem ? How many technical replicates ? How many biological replicates ? Have these results been reproduced ever ? If yes, how many times ? All knock-downs are performed with a single shRNA sequence, which is not at all the standard. At least two independent sequences should give the same phenotype before a molecule can be demonstrated to have a role.

To anticipate an easy rebuttal : some results are antagonistic between shRNA and inhibitors with IL6 or IL8 also having a role at low cell density. So inhibitors cannot simply be used as independent confirmation of the shRNA effects. ShRNA results should be ascertained with two sequences. Sigma sells several sequences for each gene. This is straightforward to validate the presented results.

We apologize for not including this critical piece of information. All the data in the paper is represented as mean \pm s.e.m. *P<0.05; **P<0.01; ***P<0.001(ANOVA). The migration, proliferation, PCR, and ELISA data was obtained from three independent experiments with at least two technical repeats. All this information has now been included in the figure captions and methods section.

Following the reviewer’s recommendation we performed experiments with an additional shRNA sequence. Our data shows there is no significant difference between cell speed between the two sequences for each knockdown in high density and low density conditions.

IL-6 sh59205	CCGGCATCTCATTCTGCGCAGCTTTCTCGAGAAAGCTGCGCAGAATGAGATGTTTTTG
IL-6 sh59203	CCGGCTGGATTCAATGAGGAGACTTCTCGAGAAGTCTCCTCATTGAATCCAGTTTTTG
IL-8 sh232053	CCGGTGCGCCAACACAGAAATTATTCTCGAGAATAATTTCTGTGTTGGCGCATTTTTTG
IL-8 sh58030	CCGGCAAGGAGTGCTAAAGAACTTACTCGAGTAAGTTCTTTAGCACTCCTTGTTTTTG

5_RNAseq : The authors claim that the main result of this analysis is that medium containing both IL6 and IL8 at the right stoichiometry recapitulates the high density medium. This is a very important point of the demonstration. But is it really the case ? The principal component analysis of results does not clearly indicate that HD and RM conditions are clustered together. HD is closer to the LD, IL6, IL8 conditions than to the RM with which the authors try to make it fit. There should be an objective score associated with this PCA. The visual impression on two clusters of selected genes does not suffice to demonstrate this point. Nowhere can I find details as to how these two clusters were defined. Nor what these genes having a biological function in ‘cellular movements’ are. Are WASF3 and Arp2/3 gene among them ? The RNAseq data should be deposited in an appropriate database (GEO) and the results concerning the clusters should be made available as supplementary data.

I guess WASF3 and Arp2/3 were simply selected as good candidates based on the extensive literature implicating them in migration. But the writing is ambiguous on this point. I also disagree with the statement that the RNAseq analysis suggests that RM induced a reduced cell metabolic activity, cell cycle and division. If this point is of importance for the manuscript, it should be directly demonstrated. Formulated as it is, it is just to mystify the reader to make him believe that migration should oppose proliferation (see first point).

Thank you for the comments and the recognition that synergistic paracrine signaling is a very important main point of this manuscript.

First, principal component analysis reveals the direction of phenotypic shift of high-density cells (HD) is toward the IL6/IL8 treated cells (RM), in contrast to low density (LD) as well as cells treated by IL6 or IL8 only. We appreciate it that the reviewer raised a good question regarding the relative distance between HD-RM distance and RM-LD distance in PCA. We believe that is due in part to the stimulation strength/dosage. The current IL6/IL8 treatment condition is strong and the treatment with reduced cytokine concentrations may bring the transcriptional phenotype of RM closer to HD. What is most important is that there exists a distinct phenotypic shift from the LD, IL8-alone, IL6-alone cluster to either HD or RM, which is from the major principal component 1 (PC1) positive to PC1 negative. Both HD and RM reside in the same quadrant of

the PCA plot, suggesting their phenotypic similarity despite the varying levels of gene expression between HD and RM.

Second, there should be differences between HD and RM in theory, and more detailed gene expression and pathway analysis (gene ontology) results are included in revised manuscript -- Supplementary Figure 4 A-B. Gene expression analysis with RNA-seq was conducted across the whole genome and clustered with top ranked differentially expressed genes in an unbiased manner. The clustering analysis indicates LD, IL6, IL8 samples are essentially identical in transcriptional profile but surprisingly synergistic IL6/IL8 treatment led to a drastic change in gene expression, in particular, RM shows upregulation of genes in clusters 3 and 4 associated with cell movement and cell-cell signaling and downregulation of genes in cluster 2 associated with cell death. HD recapitulates part of the RM transcriptional program including most genes in cluster 2 and a subset of genes in cluster 3. The major differences between RM and HD are the presence of cells initiating death programs (cluster 4) due to the harsh condition used to dissociate 3D matrix to prepare single cell suspension for RNA-seq. The pathway analysis results are summarized in Fig 4B. The top ranked pathways are mostly related to cell migration, cell-to-cell signaling, metabolic activity, and cellular assembly and organization, which are all highly relevant to the proposed mechanism. Although this IPA analysis can inform the molecular pathways associated with transcriptional phenotype change, it does not determine if downregulation of a gene cluster leads to decrease of associated biological functions or canonical pathways. Therefore, we have revised the manuscript regarding the statement “**RM induced a reduced cell metabolic activity, cell cycle and division**”. Thank you for your valuable feedback. (Supplementary figure 4A and B)

6_Mechanism p10 : JAK2, STAT3, Arp2/3 inhibitors do not only affect density induced migration but also basal migration. This is absolutely expected for the Arp2/3 complex, which is well established to be required for protrusion based migration. Actually for me, the data with CK666 should be associated with Fig1F and G about the basic characterization of migration in this system. The fact that JAK2 and STAT3 are required for basal migration is puzzling to me. How do the author interpret these results ? Do these results actually confirm the role of IL6 and IL8 in density induced migration ? I found them a bit problematic.

JAK/STAT is a central axis of cytokine signaling that controls many signal transduction pathways including the ones regulating proliferation, migration, and apoptosis. STAT3 is a master transcription factor downstream of IL-6 receptor and its activation may lead to the production of other proteins that are stimulatory, pro-growth or chemoattractant, which is not studied in this work but might represent a mechanism for JAK/STAT to further amplify the density-induced migration phenotype.

Protrusion analysis demonstrates inhibition of JAK2 and STAT3 using the inhibitors AG-490 and S3I-201 respectively also affects basal migration through the regulation of protrusions as shown in the figure below. This result suggests that JAK2 and STAT3 play a role in the regulation of basal migration through protrusion dynamics through either the actin or microtubule cytoskeleton.

The data obtained using the JAK2 and STAT3 inhibitors gave us an indication that these intermediates are involved in cell-density dependent migration. The upregulated expression of phosphoSTAT3 in high density conditions confirms its involvement in cell density dependent migration that is induced through the synergistic paracrine signaling of IL-6 and IL-8.

We thank the reviewer for the constructive criticism and have accordingly added the figure below (Supplementary figure 4D) and relevant discussion to the revised manuscript.

1. Ng, D. C. H., Lin, B. H., Lim, C. P., Huang, G., Zhang, T., Poli, V., & Cao, X. (2006). Stat3 regulates microtubules by antagonizing the depolymerization activity of stathmin. *The Journal of cell biology*, 172(2), 245-257.

7_WASF3 is induced at the mRNA level. What about the protein ? WASF3 has been found to be critical for invasion by the groups of Sossey-Alaoui and Cowell and dispensable by Machesky's group, in the same cell line MDA-MB-231 used here. Since the authors went to suggest that WASF3 is part of the mechanism of migration and invasion, they should demonstrate that WASF3 is indeed required through shRNA mediated depletion.

As mentioned by the reviewer, WASF3 has been shown to play a critical role in tumor cell migration and thus, metastasis. ShRNA mediated depletion of WASF3 would only confirm what is already known about the role of WASF3 in cell migration and would not effectively show its role in cell-density dependent migration induced by paracrine signaling. Following the reviewer's suggestion, to further confirm its role as an important intermediate in the synergistic paracrine signaling pathway that regulates cell-density dependent migration, we show that the expression of WASF3 is upregulated under a high density condition. The protein data (western blot) and the relevant references (see below) have been included in the revised manuscript. (Supplementary figure 4F)

1. Teng, Y., Ghoshal, P., Ngoka, L., Mei, Y., & Cowell, J. K. (2013). Critical role of the WASF3 gene in JAK2/STAT3 regulation of cancer cell motility. *Carcinogenesis*, bgt167.
2. Teng, Y., Ngoka, L., Mei, Y., Lesoon, L., & Cowell, J. K. (2012). HSP90 and HSP70 proteins are essential for stabilization and activation of WASF3 metastasis-promoting protein. *Journal of Biological Chemistry*, 287(13), 10051-10059.
3. Teng, Yong, et al. "Inactivation of the WASF3 gene in prostate cancer cells leads to suppression of tumorigenicity and metastases." *British journal of cancer* 103.7 (2010): 1066-1075.

8_Arp2/3. The Arp2/3 complex is a complex. Here one subunit only is measured at the mRNA level and conclusion is drawn for the whole complex. We don't even know which subunit is measured. On what criterium has this subunit been selected ? Is the level of the chosen subunit reflecting the level of others ? There are also paralogous subunits in Arp2/3 complexes, with different functions (recent work from Michael Way's group). Furthermore, one wonders why the chosen Arp2/3 subunit is not measured in Fig.4 like WASF3 for its induction by IL6 and IL8. The overall Arp2/3 complex should be measured at the protein level to conclude about any putative induction.

We apologize for the lack of detail about the subunit we measured. The subunit that we measured for the complex was the ARPC2. We chose this subunit because it forms a stable dimer with the ARPC4 subunit and together these subunits form the backbone of the complex. These two dimers bind to the side of the actin filaments which contribute to F actin assembly and produce dendritic protrusions for motility. Previous studies have also shown the importance of ARP2/3 complex by inhibiting/knocking down the ARPC2 subunit. We have specified the subunit in our updated manuscript.

Following the reviewer's recommendation, we conducted a western blot to determine the expression of ARPC2 in low and high density conditions. The data demonstrates that ARPC2 is upregulated in high density conditions. We have included this data and the references below in the revised manuscript (Supplementary Figure 4F) (**New citations #51 and 52**)

1. Pollard, T. D., & Beltzner, C. C. (2002). Structure and function of the Arp2/3 complex. *Current opinion in structural biology*, 12(6), 768-774.
2. Dang, I., Gorelik, R., Sousa-Blin, C., Derivery, E., Guérin, C., Linkner, J., ... & Ermilova, V. D. (2013). Inhibitory signaling to the Arp2/3 complex steers cell migration. *Nature*, 503(7475), 281-284.

Arp2/3 staining in IHC is displayed in Fig5J. I cannot see well the staining on the figure... The positive area is nonetheless measured. Can the intensity be measured ? Is it varying ? What does the varying area means ? Intratumoral heterogeneity of samples ? If yes, what are the morphological structures that express the Arp2/3 ? The claimed data raise all these questions, but the basics are not ascertained. The exact reference of the Ab used is not given. Again, which subunit does it target ? Is it the same as the one which has been measured by

qRT-PCR ? Has the antibody been validated for IHC ? How can we be sure that it binds to the expected target ?

Kaplan Meier of public Arp2/3 data. This is a strange way to finish the paper as the implication of the Arp2/3 as prognosis factor in many cancers has been published for a long time already. None of these references are cited, they should if the authors want to make this point that the Arp2/3 is a prognosis factor. They should not pretend to discover this point on public data: Otsubo et al., Mod Pathol 2004 ; Semba, Clin Can Res 2006, Iwaya et al. Mod Pathol 2007 ; Iwaya et al, Cancer Sci 2007 ; Zheng et al, Anticancer Res. 2008 ; Kashani-Sabet et al PNAS 2009 ; Liu et al, Oncol Report 2013.

We apologize for any confusion with the methods we employed. IHC staining was quantified using ImageJ as highlighted in the article by Ruifrok, A. C., & Johnston, D. A. (2001). The intensity of the stain is correlated with the expression of the protein which in this case is the ARPC2 protein stained with the validated antibody from proteintech (10922-1-AP). The differences in the intensity of the staining may not be apparent to the untrained eye which is why the ImageJ analysis is conventionally used to quantify staining. The ARPC2 subunit was the same subunit that was measured using qRT-PCR. Following the reviewer's recommendation, we have included this information in the manuscript.

Below are high magnification images of the IHC staining of tumors from the control group and the combination group (treated with both Tocilizumab and Reparixin). The staining demonstrates that ARPC2 is found within the cell body (staining is seen around the nucleus).

We thank the reviewer for the constructive feedback. First, we apologize for not correctly citing the works of those who helped produce the data for the Kaplan Meier data. We also apologize if the reviewer interpreted that we discovered this point. Our intention with this panel was to further strengthen the point that ARP2/3 is a good prognostic factor. We agree with the reviewer that the results section of the paper should have ended differently. We have revised the manuscript with a better ending for the results section and added the references that were suggested by the reviewer. (New citations #40, #41, #42, #43, #44, #70)

CONTROL

COMBINATION

1. Ruifrok, A. C., & Johnston, D. A. (2001). Quantification of histochemical staining by color deconvolution. *Analytical and quantitative cytology and histology*, 23(4), 291-299.

9_ambiguities:

P5 l 107-108 : for 16.5 h or for 5 days ?

We apologize for any confusion, the experiment entailed allowing the cells to grow and migrate in the matrices for 5 days and images of the cells were taken every other day for 16.5 hours to determine cell speed. We adjusted the phrasing in the manuscript to make this clearer to the reader.

P8 l189 : shRNA induction ? I did not see that shRNA treatments were induced...

We apologize for this mistake by the author. We have corrected this error in the manuscript.

Reviewers' comments:

Reviewer #1 (Remarks to the Author):

The authors have satisfactorily addressed my concerns.

Reviewer #2 (Remarks to the Author):

I went quickly through the revised manuscript and the rebuttal letter. I have the feeling that the authors only superficially revised their work.

They pretended to have done it by adding requested experiments. But: In many instances, the answers don't really match the concerns I have raised. There is no need to thank me at each answer. Just make sure that each sentence of the answer has an actual meaning. The experiments displayed are not at the level of the standard in the field (migration analysis). They neglect the fact that some proteins they study are well known to be part of complexes. Many references are obviously not at the right place. Even when I don't know them, I could notice it, just by looking at the titles of these references !

So, new experiments set aside - which I did not carefully examine yet - I have the uneasy feeling that I gave more care to my review than the authors gave to their rebuttal. There are 3 senior authors to this study, who will ultimately be responsible for the published content. I certainly don't plan to do their job and try to raise the level of the manuscript myself. So please provide me with a carefully revised version and I will carefully re-review.

Response to reviewers' comments

Reviewer #1 (Remarks to the Author):

The authors have satisfactorily addressed my concerns.

Reviewer #2 (Remarks to the Author):

I went quickly through the revised manuscript and the rebuttal letter. I have the feeling that the authors only superficially revised their work.

They pretended to have done it by adding requested experiments. But: In many instances, the answers don't really match the concerns I have raised. There is no need to thank me at each answer. Just make sure that each sentence of the answer has an actual meaning. The experiments displayed are not at the level of the standard in the field (migration analysis). They neglect the fact that some proteins they study are well known to be part of complexes. Many references are obviously not at the right place. Even when I don't know them, I could notice it, just by looking at the titles for these references !

So, new experiments set aside - which I did not carefully examine yet - I have the uneasy feeling that I gave more care to my review than the authors gave to their rebuttal. There are 3 senior authors to this study, who will ultimately be responsible for the published content. I certainly don't plan to do their job and try to raise the level of the manuscript myself. So please provide me with a carefully revised version and I will carefully re-review.

Following the reviewer's comments after the first resubmission, we knockdown WASF3 through shRNA mediated depletion and determined that it played a critical role in cell density-dependent migration. Further we conducted a western blot to determine the expression of ACTR2 and ACTR3 in low and high density conditions and found that these two proteins were upregulated at high density condition and thus played an important role in cell density-dependent migration. We also apologize for the error in our first resubmission where we mistakenly referred to ACTR2 as ARPC2.

In this manuscript by Jayatilaka et al., it is shown that sarcoma and carcinoma cells grown at high density in collagen gels display enhanced migration. This phenomenon requires IL6 and IL8 paracrine signaling. The importance of these findings is validated in xenograft assays, where inhibitors of IL6 and IL8 receptors decrease the metastatic burden. The overall message is important as it points to a potential treatment of metastasis formation. The core of the demonstration centered on IL6 and IL8 is well demonstrated. There are already multiple pieces of evidence in the literature that IL6 and IL8 are involved in cancer progression and metastasis formation. But the way these factors were found in a logical scientific and unbiased line here makes this study interesting. However, some other parts, especially the one dealing with WASF3 and Arp2/3 are simply, of poor quality. I believe major revisions are required before this manuscript can be published in Nature communications.

1_I strongly disagree with the way the study is introduced in the abstract and introduction, by stating that proliferation and migration are mutually exclusive. The authors can give a couple of references in support of this claim. But just as many can be given in support of the opposite claim. The authors should think of growth factors that trigger motility AND proliferation. HGF/SF is probably the best example but this is actually a general scheme and it works the same with EGF or PDGF. Another example at the level of the Arp2/3 complex is the Arp2/3 inhibitory

protein Arpin negatively regulate migration and proliferation in a coordinated manner. So the abstract statement that a direct coupling between migration and proliferation had never been observed is simply shocking. There is no need to build this artificial opposition between proliferation and migration to enter into this study.

We apologize for the lack of clarity when describing proliferation and migration as mutually exclusive. The concept of historic “go-or-grow” hypothesis is based on the observation of cell behavior at a given time point (see references 1-6 below). In a prolonged timeframe, a cell may take a sequence of events to realize proliferation and migration/spreading. In addition, as the reviewer points out, it is often required physiologically since an external cue such as growth factor can trigger both motility AND proliferation. Although the “go-or-grow” concept is still valid; when the cell divides it does not migrate and when it migrates it is not dividing. The key finding of this work is that as cells proliferate and **local cell density increases**, the effect of intercellular paracrine signaling is enhanced, leading to a change in the secretion profile and thus enhancement of cell migration, which is implicated in tumor invasion and metastasis. We believe the way of coupling of cancer cell proliferation and migration through cell density discovered in this work is novel. We have made changes to the manuscript to convey this message more clearly. We thank the reviewer for the valuable feedback. **(New citations #4, #5, and #10)**

1. De Donatis, A., Ranaldi, F., & Cirri, P. (2010). Reciprocal control of cell proliferation and migration. *Cell Communication and Signaling*, 8(1), 20.
2. Evdokimova, V., Tognon, C., Ng, T., & Sorensen, P. H. Reduced proliferation and enhanced migration: two sides of the same coin? Molecular mechanisms of metastatic progression by YB-1. *Cell cycle*, 8(18), 2901-2906, (2009)
3. Palmqvist, R., Rutegård, J. N., Bozoky, B., Landberg, G., & Stenling, R., Human colorectal cancers with an intact p16/cyclin D1/pRb pathway have up-regulated p16 expression and decreased proliferation in small invasive tumor clusters. *The American journal of pathology*, 157(6), 1947-1953, (2000)
4. Svensson, S., Nilsson, K., Ringberg, A., & Landberg, G., Invade or proliferate? Two contrasting events in malignant behavior governed by p16INK4a and an intact Rb pathway illustrated by a model system of basal cell carcinoma. *Cancer research*, 63(8), 1737-1742. (2003).
5. Hoek, K. S., Eichhoff, O. M., Schlegel, N. C., Döbbeling, U., Kobert, N., Schaerer, L., ... & Dummer, R., In vivo switching of human melanoma cells between proliferative and invasive states. *Cancer research*, 68(3), 650-656, (2008).
6. Zheng, P. P., Severijnen, L. A., van der Weiden, M., Willemsen, R., & Kros, J. M. (2009). Cell proliferation and migration are mutually exclusive cellular phenomena in vivo: implications for cancer therapeutic strategies. *Cell Cycle*, 8(6), 950-951.

2_I was expecting some discussion of how cell density is sensed in mammalian cells. It should be discussed and if there are some easy ways to show that the current observations are in a classical framework or on the contrary in an atypical one, they should also be performed. Paracrine signaling in response to cell density is the core of the paper.

The cell density sensing is presumably via paracrine signaling through IL-6 receptor (IL6R) and IL-8 receptor (IL8R1/CXCR1 or IL8R2/CXCR2). In a 3D type I collagen matrix a gradient of secretion proteins can readily build up around a cell and paracrine signaling can occur when the inter-cellular distance is decreased with the increase of cell density. As a result, the paracrine signaling could trigger a response of cellular behavior (e.g., enhanced migration) or the production of a second wave of cytokines to modulate cell density. This is the mechanism we believe how cell density is sensed in our study (mammalian cells).

This work focuses on the former effect – paracrine-induced cell behavior change (migration) and a systematic study was performed to measure the consequence of paracrine-induced intracellular signaling cascades linked to cytoskeleton, cell adhesion, and migration. In this work, we have measured both IL-6/IL-8 secretion and the expression of IL6R and IL8R (CXCR2) (See figure below), together with the downstream intracellular pathways leading to cell migration enhancement, which is a relatively complete study in the former aspect.

The latter is a complex systems biology and cell signaling research project similar as one of our recent publications (Xue et al., *Science Signaling*, 8(381):ra59, 2015) and is indeed one of our future directions. We have included the data below to the revised manuscript and added more discussion on these aspects. (Supplementary figure 3L,M)

1. Xue, Q., Lu, Y., Eisele, M.R., Sulistijo, E.S., Khan, N., Fan, R., Miller-Jensen, K., (2015). Analysis of single-cell cytokine secretion reveals a role for paracrine signaling in coordinating macrophage responses to TLR4 stimulation. *Science Signaling*, 8(381):ra59

3_Migration is not fully analyzed. Just speed is displayed. From the same trajectories the authors should extract the overall migration efficiency reflected in MSD and the directional persistence which together with speed determines the overall migration efficiency.

The cell speeds shown in the paper were obtained calculating the MSDs and then extracting the speed from it. Therefore cell speeds directly correlate to the MSDs. (Refer figure below) Following the reviewer's recommendation, we looked at persistence distance and invasive distance (as highlighted by Wu et al) of the matrix embedded cells with increasing cell density and found that the cells are more persistent at a high density than a low density. (Refer figure below) Following the reviewer's recommendation, this data was included in the revised manuscript. (Figure 1F and Supplementary Figure 1A and B) (New citations #30).

1. Wu, P. H., Giri, A., Sun, S. X., & Wirtz, D. (2014). Three-dimensional cell migration does not follow a random walk. *Proceedings of the National Academy of Sciences*, 111(11), 3949-3954.
2. Wu, Pei-Hsun, Anjil Giri, and Denis Wirtz. "Statistical analysis of cell migration in 3D using the anisotropic persistent random walk model." *Nature protocols* 10.3 (2015): 517-527.

4_Experiments are displayed with error bars. We are never really explained what is plotted. Mean + or - sem ? How many technical replicates ? How many biological replicates ? Have these results been reproduced ever ? If yes, how many times ? All knock-downs are performed

with a single shRNA sequence, which is not at all the standard. At least two independent sequences should give the same phenotype before a molecule can be demonstrated to have a role.

To anticipate an easy rebuttal : some results are antagonistic between shRNA and inhibitors with IL6 or IL8 also having a role at low cell density. So inhibitors cannot simply be used as independent confirmation of the shRNA effects. ShRNA results should be ascertained with two sequences. Sigma sells several sequences for each gene. This is straightforward to validate the presented results.

We apologize for not including this critical piece of information. All the data in the paper is represented as mean \pm s.e.m. *P<0.05; **P<0.01; ***P<0.001(ANOVA). The migration, proliferation, PCR, and ELISA data was obtained from three independent experiments with at least two technical repeats. All this information has now been included in the figure captions and methods section.

Following the reviewer's recommendation we performed experiments with an additional shRNA sequence. Our data shows there is no significant difference between cell speed between the two sequences for each knockdown in high density and low density conditions.

IL-6 sh59205	CCGGCATCTCATTCTGCGCAGCTTTCTCGAGAAAGCTGCGCAGAATGAGATGTTTTTG
IL-6 sh59203	CCGGCTGGATTCAATGAGGAGACTTCTCGAGAAGTCTCCTCATTGAATCCAGTTTTTG
IL-8 sh232053	CCGGTGCCCAACACAGAAATTATTCTCGAGAATAATTTCTGTGTTGGCGCATTTTTTG
IL-8 sh58030	CCGGCAAGGAGTGCTAAAGAACTTACTCGAGTAAGTTCTTTAGCACTCCTTGTTTTTG

5_RNAseq : The authors claim that the main result of this analysis is that medium containing both IL6 and IL8 at the right stoichiometry recapitulates the high density medium. This is a very important point of the demonstration. But is it really the case ? The principal component analysis of results does not clearly indicate that HD and RM conditions are clustered together. HD is closer to the LD, IL6, IL8 conditions than to the RM with which the authors try to make it fit. There should be an objective score associated with this PCA. The visual impression on two clusters of selected genes does not suffice to demonstrate this point. Nowhere can I find details as to how these two clusters were defined. Nor what these genes having a biological function in ‘cellular movements’ are. Are WASF3 and Arp2/3 gene among them ? The RNAseq data should be deposited in an appropriate database (GEO) and the results concerning the clusters should be made available as supplementary data.

I guess WASF3 and Arp2/3 were simply selected as good candidates based on the extensive literature implicating them in migration. But the writing is ambiguous on this point. I also disagree with the statement that the RNAseq analysis suggests that RM induced a reduced cell metabolic activity, cell cycle and division. If this point is of importance for the manuscript, it should be directly demonstrated. Formulated as it is, it is just to mystify the reader to make him believe that migration should oppose proliferation (see first point).

Thank you for the comments and the recognition that synergistic paracrine signaling is a very important main point of this manuscript.

First, principal component analysis reveals the direction of phenotypic shift of high-density cells (HD) is toward the IL6/IL8 treated cells (RM), in contrast to low density (LD) as well as cells treated by IL6 or IL8 only. We quantified the relative distances between HD-RM and RM-LD in PCA for the top 930 genes and have displayed this data in the table below. This data demonstrates that there is distinct phenotypic shift from the LD, IL8-alone, IL6-alone cluster to either HD or RM. Both HD and RM reside in the same quadrant of the PCA plot, suggesting their phenotypic similarity despite the varying levels of gene expression between HD and RM.

Second, there should be differences between HD and RM in theory, and more detailed gene expression and pathway analysis (gene oncology) results are included in revised manuscript -- Supplementary Figure 4 A-B. Gene expression analysis with RNA-seq was conducted cross the whole genome and clustered with top ranked differentially expressed genes in an unbiased manner. The clustering analysis indicates LD, IL6, IL8 samples are essentially identical in transcriptional profile but surprisingly synergistic IL6/IL8 treatment led to a drastic change in gene expression, in particular, RM shows upregulation of genes in clusters 3 and 4 associated with cell movement and cell-cell signaling and downregulation of genes in cluster 2 associated with cell death. HD recapitulates part of the RM transcriptional program including most genes in cluster 2 and a subset of genes in cluster 3. The major differences between RM and HD are the presence of cells initiating death programs (cluster 4) due to the harsh condition used to dissociate 3D matrix to prepare single cell suspension for RNA-seq. The pathway analysis results are summarized in Fig 4B. The top ranked pathways are mostly related to cell migration, cell-to-cell signaling, metabolic activity, and cellular assembly and organization, which are all highly relevant to the proposed mechanism. Although this IPA analysis can inform the molecular pathways associated with transcriptional phenotype change, it does not determine if downregulation of a gene cluster leads to decrease of associated biological functions or canonical pathways. Therefore, we have revised the manuscript regarding the statement “**RM induced a reduced cell metabolic activity, cell cycle and division**”. Thank you for your valuable feedback. (Supplementary figure 4A and B)

6_Mechanism p10 : JAK2, STAT3, Arp2/3 inhibitors do not only affect density induced migration but also basal migration. This is absolutely expected for the Arp2/3 complex, which is well established to be required for protrusion based migration. Actually for me, the data with CK666 should be associated with Fig1F and G about the basic characterization of migration in this system. The fact that JAK2 and STAT3 are required for basal migration is puzzling to me. How do the author interpret these results ? Do these results actually confirm the role of IL6 and IL8 in density induced migration ? I found them a bit problematic.

JAK/STAT is a central axis of cytokine signaling that controls many signal transduction pathways including the ones regulating proliferation, migration, and apoptosis. STAT3 is a master transcription factor downstream of IL-6 receptor and its activation may lead to the production of other proteins that are stimulatory, pro-growth or chemoattractant, which is not studied in this work but might represent a mechanism for JAK/STAT to further amplify the density-induced migration phenotype.

Protrusion analysis demonstrates inhibition of JAK2 and STAT3 using the inhibitors AG-490 and S3I-201 respectively also affects basal migration through the regulation of protrusions as shown in the figure below. This result suggests that JAK2 and STAT3 play a role in the regulation of basal migration through protrusion dynamics through either the actin or microtubule cytoskeleton.

The data obtained using the JAK2 and STAT3 inhibitors gave us an indication that these intermediates are involved in cell-density dependent migration. The upregulated expression of phosphoSTAT3 in high density conditions confirms its involvement in cell density dependent migration that is induced through the synergistic paracrine signaling of IL-6 and IL-8.

We have added the figure below (Supplementary figure 4D) and relevant discussion to the revised manuscript.

- Ng, D. C. H., Lin, B. H., Lim, C. P., Huang, G., Zhang, T., Poli, V., & Cao, X. (2006). Stat3 regulates microtubules by antagonizing the depolymerization activity of stathmin. *The Journal of cell biology*, 172(2), 245-257.

7_ WASF3 is induced at the mRNA level. What about the protein ? WASF3 has been found to be critical for invasion by the groups of Sossey-Alaoui and Cowell and dispensable by Machesky's group, in the same cell line MDA-MB-231 used here. Since the authors went to suggest that WASF3 is part of the mechanism of migration and invasion, they should demonstrate that WASF3 is indeed required through shRNA mediated depletion.

Following the reviewer's recommendation, we knockdown WASF3 in the HT1080 fibrosarcoma cells that were used to determine the pathway, through shRNA mediated depletion and observed cell speed at both low and high density. Cell density-dependent migration was not observed in WASF3 depleted cells as there is no difference in cell speeds at a low density and a high density, indicating that WASF3 is indeed critical for cell density- dependent migration

WASF3 344184	CCGGCGCTGCTATTTCGAATGGGAATCTCGAGATTCCCATTTCGAATAGCAGCGTTTTTG
WASF3 381495	GTACCGGGCATCGGACGTTACGGATTACCTCGAGGTAATCCGTAACGTCCGATGCTTTTTTG

8_Arp2/3. The Arp2/3 complex is a complex. Here one subunit only is measured at the mRNA level and conclusion is drawn for the whole complex. We don't even know which subunit is measured. On what criterium has this subunit been selected ? Is the level of the chosen subunit reflecting the level of others ? There are also paralogous subunits in Arp2/3 complexes, with different functions (recent work from Michael Way's group). Furthermore, one wonders why the chosen Arp2/3 subunit is not measured in Fig.4 like WASF3 for its induction by IL6 and IL8. The overall Arp2/3 complex should be measured at the protein level to conclude about any putative induction.

We apologize for the lack of detail about the subunit we measured. The subunit that we measured for the complex was ACTR2. Following the reviewer's recommendation, we conducted a western blot to determine the protein expression of ACTR2 and ACTR3 in low and high density conditions. The data demonstrates that both ACTR 2 and ACTR3 are upregulated in high density conditions further confirming that ARP2/3 is important in regulating cell density-dependent migration

Arp2/3 staining in IHC is displayed in Fig5J. I cannot see well the staining on the figure... The positive area is nonetheless measured. Can the intensity be measured ? Is it varying ? What does the varying area means ? Intratumoral heterogeneity of samples ? If yes, what are the morphological structures that express the Arp2/3 ? The claimed data raise all these questions, but the basics are not ascertained. The exact reference of the Ab used is not given. Again, which subunit does it target ? Is it the same as the one which has been measured by qRT-PCR ? Has the antibody been validated for IHC ? How can we be sure that it binds to the expected target ? Kaplan Meier of public Arp2/3 data. This is a strange way to finish the paper as the implication of the Arp2/3 as prognosis factor in many cancers has been published for a long time already. None of these references are cited, they should if the authors want to make this point that the Arp2/3 is a prognosis factor. They should not pretend to discover this point on public data: Otsubo et al., Mod Pathol 2004 ; Semba, Clin Can Res 2006, Iwaya et al. Mod Pathol 2007 ;

Iwaya et al, Cancer Sci 2007 ; Zheng et al, Anticancer Res. 2008 ; Kashani-Sabet et al PNAS 2009 ; Liu et al, Oncol Report 2013.

We apologize for any confusion with the methods we employed. IHC staining was quantified using ImageJ as highlighted in the article by Ruifrok, A. C., & Johnston, D. A. (2001). The intensity of the stain is correlated with the expression of the protein which in this case is the ACTR2 protein stained with the validated antibody from proteintech (10922-1-AP). The differences in the intensity of the staining may not be apparent to the untrained eye which is why the ImageJ analysis is conventionally used to quantify staining. The ACTR2 subunit was the same subunit that was measured using qRT-PCR. Following the reviewer's recommendation, we have included this information in the manuscript.

Below are high magnification images of the IHC staining of tumors from the control group and the combination group (treated with both Tocilizumab and Reparixin). The staining demonstrates that ACTR2 is found within the cell body (staining is seen around the nucleus).

We thank the reviewer for the constructive feedback. First, we apologize for not correctly citing the works of those who helped produce the data for the Kaplan Meier data. We also apologize if the reviewer interpreted that we discovered this point. Our intention with this panel was to further strengthen the point that ARP2/3 is a good prognostic factor. We agree with the reviewer that the results section of the paper should have ended differently. We have revised the manuscript with a better ending for the results section and added the references that were suggested by the reviewer. (New citations #40, #41, #42, #43, #44, #70)

CONTROL

COMBINATION

1. Ruifrok, A. C., & Johnston, D. A. (2001). Quantification of histochemical staining by color deconvolution. *Analytical and quantitative cytology and histology*, 23(4), 291-299.

9_ambiguities:

P5 l 107-108 : for 16.5 h or for 5 days ?

We apologize for any confusion, the experiment entailed allowing the cells to grow and migrate in the matrices for 5 days and images of the cells were taken every other day for 16.5 hours to determine cell speed. We adjusted the phrasing in the manuscript to make this clearer to the reader.

P8 l189 : shRNA induction ? I did not see that shRNA treatments were induced...

We apologize for this mistake by the author. We have corrected this error in the manuscript

REVIEWERS' COMMENTS:

Reviewer #2 (Remarks to the Author):

In its revised form, the work is now acceptable for publication.